**Data Availability Statement:** All relevant data are within the paper and its Supporting Information files.

# The evolutionarily conserved PhLP3 is essential for sperm development in *Drosophila melanogaster*

Christopher Petit[1], Elizabeth Kojak[1¤a], Samantha Webster📷[1], Michela Marra[2], Brendan Sweeney📷[1¤b], Claire Chaikin📷[1¤c], Jennifer C. Jemc[1,3‡]*, Stefan M. Kanzok📷[1,3‡]*

**1** Department of Biology, Loyola University Chicago, Chicago, Illinois, United States of America, **2** Department of Microbiology and Immunology, Stritch School of Medicine, Loyola University Chicago, Chicago, Illinois, United States of America, **3** Bioinformatics Program, Loyola University Chicago, Chicago, Illinois, United States of America

¤a Current address: Lake Erie College of Osteopathic Medicine, Erie, Pennsylvania, United States of America
¤b Current address: George Washington University School of Medicine & Health Sciences, Washington, DC, United States of America
¤c Current address: Driskill Life Sciences Graduate Program, Northwestern University, Chicago, Illinois, United States of America
‡ JCJ and SMK are joint senior authors on this work.
* skanzok@luc.edu (SMK); jmierisch@luc.edu (JCJ)

## Abstract

Phosducin-like proteins (PhLP) are thioredoxin domain-containing proteins that are highly conserved across unicellular and multicellular organisms. PhLP family proteins are hypothesized to function as co-chaperones in the folding of cytoskeletal proteins. Here, we present the initial molecular, biochemical, and functional characterization of *CG4511* as *Drosophila melanogaster PhLP3*. We cloned the gene into a bacterial expression vector and produced enzymatically active recombinant PhLP3, which showed similar kinetics to previously characterized orthologues. A fly strain homozygous for a P-element insertion in the 5' UTR of the *PhLP3* gene exhibited significant downregulation of *PhLP3* expression. We found these male flies to be sterile. Microscopic analysis revealed altered testes morphology and impairment of spermiogenesis, leading to a lack of mature sperm. Among the most significant observations was the lack of actin cones during sperm maturation. Excision of the P-element insertion in *PhLP3* restored male fertility, spermiogenesis, and seminal vesicle size. Given the high level of conservation of PhLP3, our data suggests PhLP3 may be an important regulator of sperm development across species.

## Introduction

The phosducin-like family of proteins (PhLP/PDCL/Plp) consists of small cytoplasmic proteins that are evolutionarily conserved among eukaryotes from yeast to humans [1–4] (Fig 1A and 1B). *Drosophila melanogaster* PhLP3 shares significant amino acid sequence identities with homologous proteins in a wide range of eukaryotic organisms (Fig 1A). Their

**Funding:** This work was supported by NSF MRI #1828164, which provided for purchasing the Zeiss LSM880 confocal microscope (JCJ and SMK). This work was furthermore supported by internal research awards from Loyola University Chicago (CP, EK, SW, MM, BS, CC, JCJ, and SMK). The funders had no role in the data collection and analysis, publication decision, or manuscript preparation. No additional external funding was received for this study.

**Competing interests:** The authors have declared that no competing interests exist.

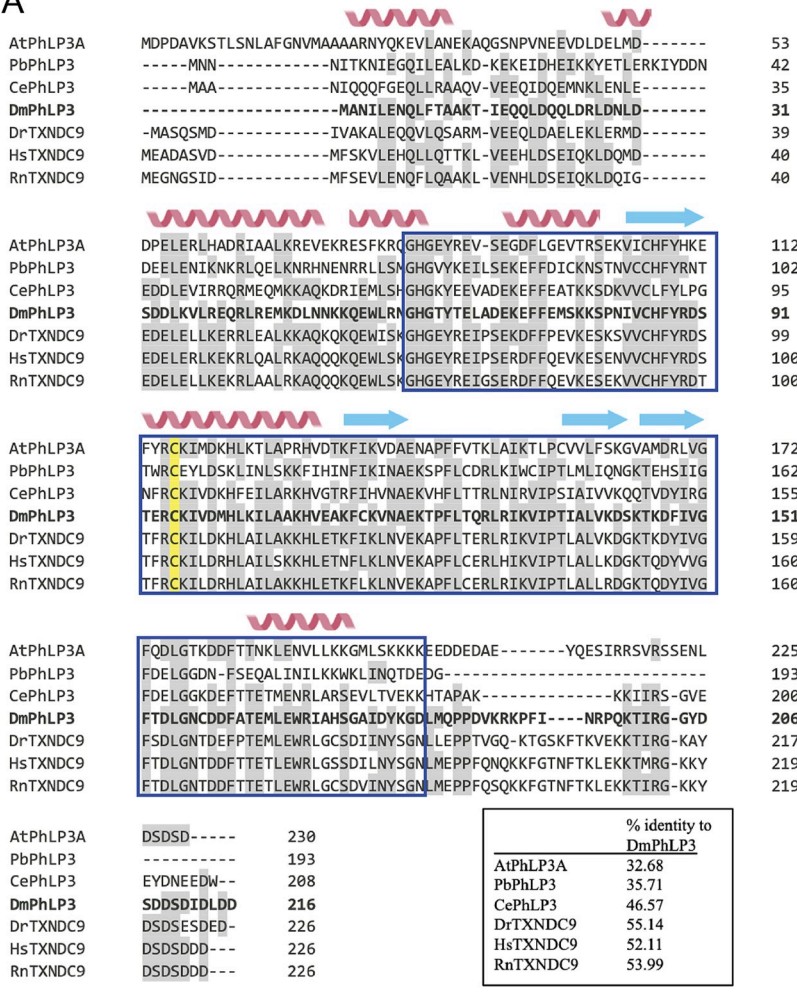

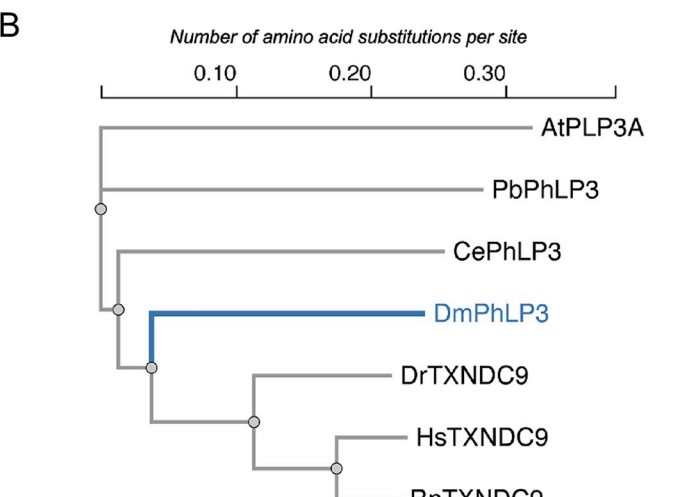

**Fig 1. Sequence and structure conservation of PhLP3 proteins.** (A) Clustal Omega alignment of published sequences. Similar amino acids are underlaid in gray. Secondary structure predictions by JPred4 [11] are shown above the sequences with alpha helices in red and beta strands in yellow. The blue box indicates the thioredoxin domain. The conserved putative redox-active cysteine is highlighted in yellow. The inset table shows sequence identities between *Drosophila melanogaster* PhLP3 and the sequences in the alignment. Abbreviations and NCBI references: At =

*Arabidopsis thaliana* (AEE78732; [12]), Pb = *Plasmodium berghei* (SCO61590; [4]), Ce = *Caenorhabditis elegans* (NP_498410; [2]), Dm = *Drosophila melanogaster* (AAL28410; this paper), Dr = *Danio rerio* (AAI65424; [13]), Hs = *Homo sapiens* (NP_005774; [4, 6]), Rn = *Rattus norwegicus* (NP_742051; [14]). (B) Phylogenetic tree of the aligned sequences. Alignment and phylogenetic tree were generated using EMBL-EBI Clustal Omega tools [15]. The scale represents the "length" of the branches, which indicates the evolutionary distance between the sequences in units of amino acid substitutions per site. Longer branches represent larger numbers of genetic changes.

hypothesized cellular functions range from regulating trimeric G-proteins and actin and tubulin-based cytoskeletal structures to chaperone-assisted protein folding [2, 5, 6]. The PhLP family is organized into three subgroups based primarily on sequence similarities and proposed cellular functions [7]. The PhLP1 group is involved in the folding, stabilizing, and regulating the beta-gamma dimers of trimeric G-proteins (reviewed in [8]). Members of the PhLP2 and PhLP3 groups are hypothesized to function as co-chaperones in conjunction with the *Chaperonin-containing tailless complex polypeptide 1* (CCT), also referred to as the *tailless complex polypeptide 1 ring complex* (TRiC). The CCT is an essential molecular cytosolic chaperonin in eukaryotes responsible for folding a variety of cytosolic proteins, including α- and β-tubulin and actin [6, 9, 10].

PhLPs have been shown to be essential for unicellular organisms, including yeast *Saccharomyces cerevisiase*, the protozoan parasite *Plasmodium berghei*, and the amoeba *Dictyostelium discoideum* [4, 7, 10]. RNAi-mediated knockdown of *PhLP* in multicellular organisms, such as the nematode *Caenorhabditis elegans*, alters cytoskeletal-dependent, including nuclear division and cytokinesis, and renders embryos unable to complete the first division [2]. Similar observations were reported in the plant *Arabidopsis thaliana*, where simultaneous knockdown of *PLP3A* and *PLP3B* disrupts microtubule-dependent oriented cell expansion in the meristem [12]. *PhLP3* silencing or overexpression in Chinese Hamster Ovary (CHO) cells leads to disorganization of microtubules as well as actin-based cytoskeletal structures [16]. Several recent cryo-electron microscopy (CryoEM) studies on the yeast and human CCTs show that PhLP, together with tubulin or actin, localizes to the central chamber of the chaperonin ring complex [17–19]. Taken together, these data suggest that the function of PhLP3 is linked to the regulation and dynamics of the cytoskeleton.

All PhLPs exhibiting molecular weights between 25 and 30 kD appear to share the same overall structural organization, consisting of an N-terminal helix domain, which ranges from 150 AA in PhLP1 to 59 AA in PhLP3, a central thioredoxin (Trx) domain comprised of 130 AA, and an unstructured C-terminal tail of varying length (Fig 1) [1]. Within the structure, the Trx-domain exhibits the highest degree of sequence conservation [7]; for example, the protozoan organism *Plasmodium berghei* and its human homolog TXNDC9 share 49% AA sequence identity between their respective Trx-domains (Fig 1) [4]. Furthermore, our group recently demonstrated that *P. berghei* PhLP3 and human TXNDC9 exhibit 1-cysteine-based redox activity [4]. These new insights into the molecular mechanism of this highly conserved protein could significantly advance our understanding of the biological role(s) of this conserved family of proteins.

Here we utilized *D. melanogaster* as a model system to further explore the function of PhLPs at a cellular and molecular level. FlyBase currently lists three genes in the *D. melanogaster* genome as potentially belonging to the PhLP family, namely *CG4511*, *CG7650*, and *viral inhibitor of apoptosis-associated factor* (*viaf*/*CG18593*) (S1 Fig) [20]. None of these genes have been experimentally characterized to date. Previous studies that included the *D. melanogaster* *CG4511* gene in multiple sequence alignments show that it aligns with the PhLP3 group [2, 7]. Therefore, we analyzed the conservation and predicted structure of the protein encoded by *CG4511* as a potential PhLP3 family member in *D. melanogaster*.

High-throughput expression data and single-cell RNA-sequencing (scRNA) data show high *CG4511* expression in the testis, specifically in germline cells (S2 Fig) [21–23]. This prompted us to examine its cellular function in the context of spermatogenesis. Spermatogenesis in *D. melanogaster* is a stepwise process in which germline stem cells (GSCs) divide to give rise to differentiating daughter cells, known as gonialblasts. Gonialblasts undergo four rounds of mitosis followed by meiosis (Fig 2; reviewed in [24]). During the post-meiotic stages of sperm development, known as spermiogenesis, spermatids extend axonemes, and nuclei begin to elongate, progressing through the leaf and canoe stages to a final needle-like morphology (Fig 2; [25]). Microtubules are critical for axoneme formation and function, and for nuclear elongation, as components of the dense complex [25]. Actin has also been observed in the dense complexes, as well as the individualization complexes that promote sperm individualization [26,

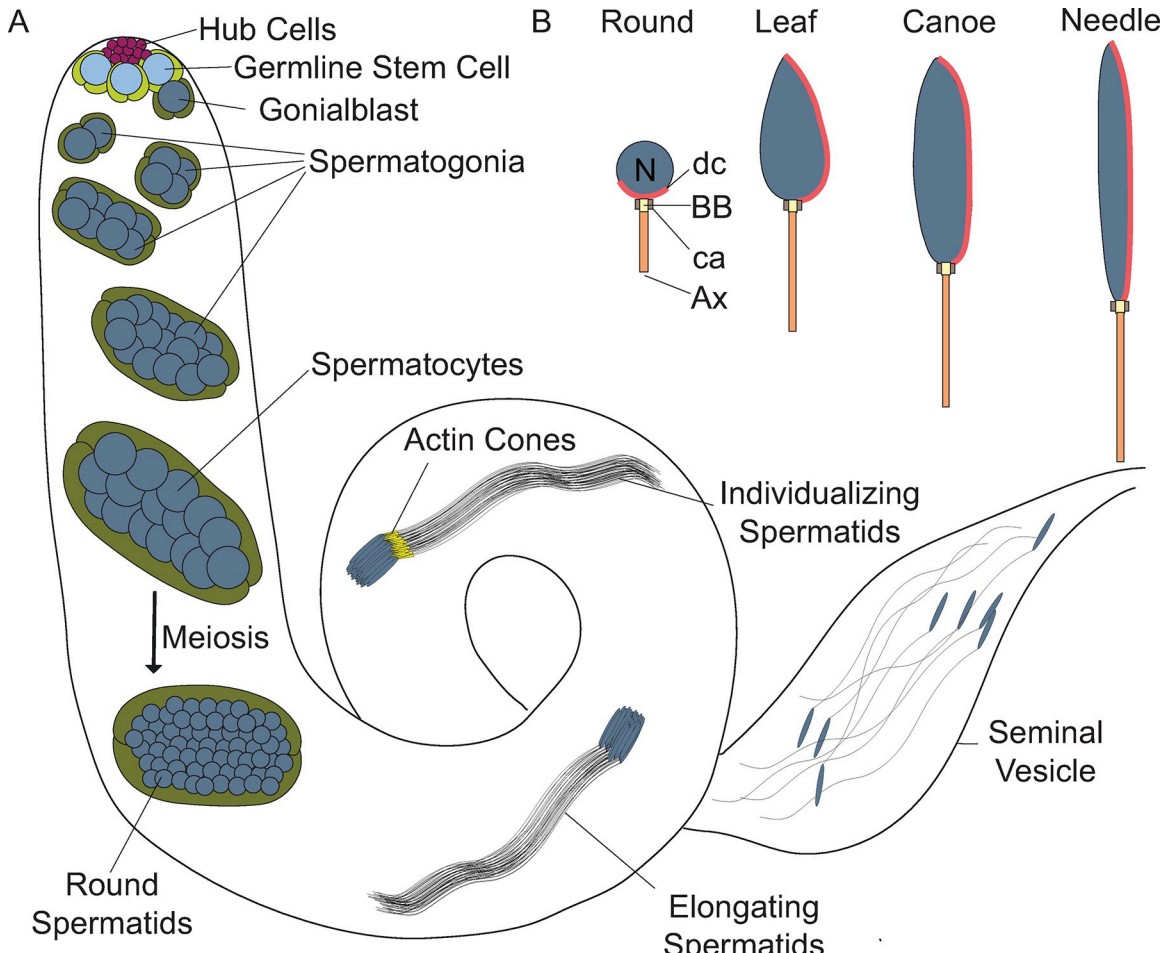

**Fig 2. *Drosophila melanogaster* sperm development.** (A) Stages of *D. melanogaster* spermatogenesis. Germline stem cells (GSCs; light blue) and cyst stem cells (CySCs; light green) are regulated and maintained by hub cells (magenta) at the apical tip. GSCs and CySCs divide asymmetrically to give rise to differentiating daughter cells, known as gonialblasts (dark blue) and somatic cyst cells (dark green), respectively. Gonialblasts undergo four mitotic divisions with incomplete cytokinesis, yielding cysts of 16 interconnected spermatogonia. Following mitosis, spermatogonia mature into spermatocytes, which grow and proceed through meiosis with incomplete cytokinesis to form 64 interconnected, haploid, round spermatids encased within a syncytial cyst. The post-meiotic maturation of spermatids is referred to as spermiogenesis. During this stage, spermatids extend axonemes, elongate nuclei, and are individualized by actin-based individualization complexes, also known as actin cones or investment cones (yellow triangles). Mature sperm coil and move into the seminal vesicle [24]. (B) Stages of nuclear elongation during spermiogenesis. Following meiosis, spermatid nuclei are round and begin to elongate, passing through the leaf and canoe stage to the final needle-like stage [25]. N: nucleus, BB: basal body, dc: dense complex, ca: centriolar adjunct, and Ax: axoneme (axoneme not to scale).

27]. During individualization, superfluous organelles and cytoplasm are stripped away, while each sperm cell is surrounded by its own plasma membrane [26]. Mature, individualized sperm coil and move into the seminal vesicle, where they are stored (Fig 2; [28]).

We have identified the *D. melanogaster* homolog of PhLP3 as the protein encoded by the previously uncharacterized *CG4511* gene. Sequence alignment and structural modeling reveal a high level of conservation, which includes a critical cysteine in the active site of the thioredoxin domain. We demonstrated that *D. melanogaster* PhLP3 is redox-active, similar to other PhLP3 family members. Expression of *PhLP3* in the testes led us to explore its role in sperm development. We have shown that PhLP3 is required for spermiogenesis in *D. melanogaster*. Males homozygous for a P-element insertion in the 5' UTR of *PhLP3* exhibited decreased *PhLP3* expression and infertility. In *PhLP3* mutants, spermatid nuclei were scattered throughout the syncytium and lacked actin-based individualization cones. Furthermore, needle-like nuclei indicative of mature sperm were absent, and seminal vesicle size was severely reduced. Germ-cell specific knockdown of *PhLP3* yielded phenotypes similar to those observed in mutants. Given the importance of microtubules and actin at multiple stages of sperm development and the proposed role of PhLP3 family proteins in regulating the cytoskeleton, the testis is an ideal tissue to explore the cellular role of PhLP3 proteins *in vivo*.

## Results

PhLP3 structure and redox activity are conserved across species

The *D. melanogaster* PhLP3 protein is encoded by *CG4511*, a single copy gene located on the right arm of chromosome 3 in the *D. melanogaster* genome and spanning a region of 1,912 bp. The gene is organized into two coding exons flanking a 199 bp intron. A non-coding exon situated 111 bp upstream of the start codon, referred to as exon 2A and exon 2B, gives rise to the two PhLP3 splice variants *CG4511-RA* and *CG4511-RB*, which differ in their respective 5' UTRs but not in their coding regions (Fig 3). The PhLP3 coding sequence (CDS) spans 651 bp and codes for a 216 AA protein with an $M_W$ of 25.11 kD. The AlphaFold database contains a predicted *D. melanogaster* PhLP3 structure (Uniprot ID Q9VGV8 [29, 30]), exhibiting the characteristic organization of the PhLP family of proteins (Fig 4A). It contains two N-terminal alpha-helices, a central Trx-domain with a central four-stranded beta-sheet, and four peripheral alpha-helices, followed by an unstructured C-terminal tail. We also generated a hypothetical model of *D. melanogaster* PhLP3 using the recently resolved CryoEM structure of human PhLP2A, which was resolved as part of the human CCT, to generate a hypothetical model of *D. melanogaster* PhLP3 (Fig 4B and 4C) [17, 31, 32]. This model exhibits the same basic PhLP organization (Fig 4B) [1]. The highly conserved globular Trx-domain exhibits four peripheral alpha helices surrounding a four-stranded twisted beta-sheet in a structural organization characteristic of the thioredoxin fold [33]. Notably, the first 18 N-terminal AA and the last 25 AA at the C-terminus were not modeled, as they were unresolved in the hPhLP2A crystal structure within the chaperonin complex.

The cytosolic thioredoxin system is central in distributing electrons to target proteins in various antioxidant and regulatory pathways. We previously demonstrated that human TXNDC9 and *P. berghei* PhLP3 exhibit cysteine-based redox activity with the thioredoxin system, suggesting that PhLPs may serve as target proteins *in vivo* [4]. The sequence alignment and structure model show that the redox-active cysteine seems to be conserved in PhLP3 homologs across diverse species, including CYS95 in *D. melanogaster* PhLP3 (Fig 1A). To test whether redox-activity is conserved in recombinant PhLP3, we cloned the CDS from *D. melanogaster* cDNA into a bacterial expression vector. We purified recombinant PhLP3 using a Ni-NTA-based column approach under denaturing conditions as described previously [4]. SDS

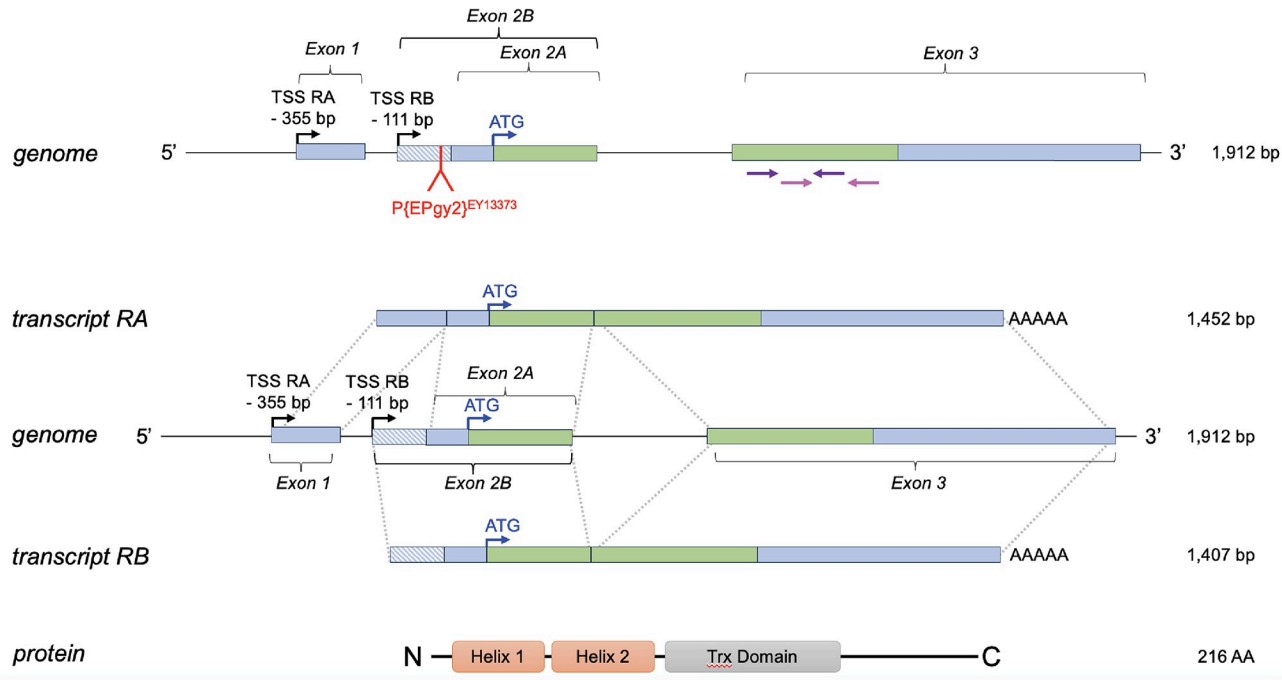

**Fig 3. Organization of the *PhLP3* gene.** The gene is located on chromosome 3 in the *D. melanogaster* genome (10854501–10856412) (FlyBase: FB2023_06, released December 12, 2023). The transcription start sites (TSS) are indicated. Intron 1 is located in the 5' UTR. Exon 2 contains a splice site resulting in two transcripts, RA and RB, containing exon 2A and exon 2B, respectively. The transcripts differ only in their 5' UTR regions and translate into identical proteins. Green boxes indicate coding regions of exons, while light blue boxes indicate untranslated regions. Protein domains are indicated according to secondary structure prediction JPred [11] as well as structural alignment of human PhLP2A using SWISS-MODEL [34]. Primers used for RT-qPCR are indicated in purple and pink.

gel electrophoresis showed a single band at ~ 25 kD, the expected size of recombinant His-tagged PhLP3 (~ 25 kD; Fig 5A). Using this method, we routinely produced >10 mg of purified recombinant PhLP3 per liter of bacterial culture.

Based on our recent findings that *P. berghei* PhLP3 and human TXNDC9 exhibit redox activity and have a conserved cysteine at position 95 (Fig 1A), we hypothesized that *D. melanogaster* PhLP3 will also be redox-active. To test this, we conducted NADPH reduction assays as outlined in Fig 5B and as described previously [4, 35]. Briefly, electron donor NADPH was added to a mix of recombinant *P. berghei* thioredoxin reductase (PbTrxR, PbANKA_0824700, [36]) and *D. melanogaster* thioredoxin (DmTrx-1, CG4193, [37]). The resulting reduction reaction was allowed to come to completion before purified recombinant PhLP3 was added. We observed renewed oxidation of NADPH, indicating the flow of electrons from the Trx-system to PhLP3 and confirming that *D. melanogaster* PhLP3 is redox-active (Fig 5C). We determined that the reaction between Trx-1 and PhLP3 follows Michaelis Menten kinetics, exhibiting a $K_m$ of 2.72 µM and a $V_{max}$ of 5.4 µM/min (Fig 5C). These values compare well with those of previously tested PhLP3 proteins, *P. berghei* PhLP3 and human TXNDC9, and support our hypothesis that redox activity is conserved in PhLPs between evolutionarily distant species and may be necessary for its biological function [4].

## PhLP3 is required for male fertility

The high expression of PhLP3 in the germ cells of *D. melanogaster* testes, as indicated by publicly available RNA-Seq and scRNA-Seq data, suggests a role for the gene in spermatogenesis

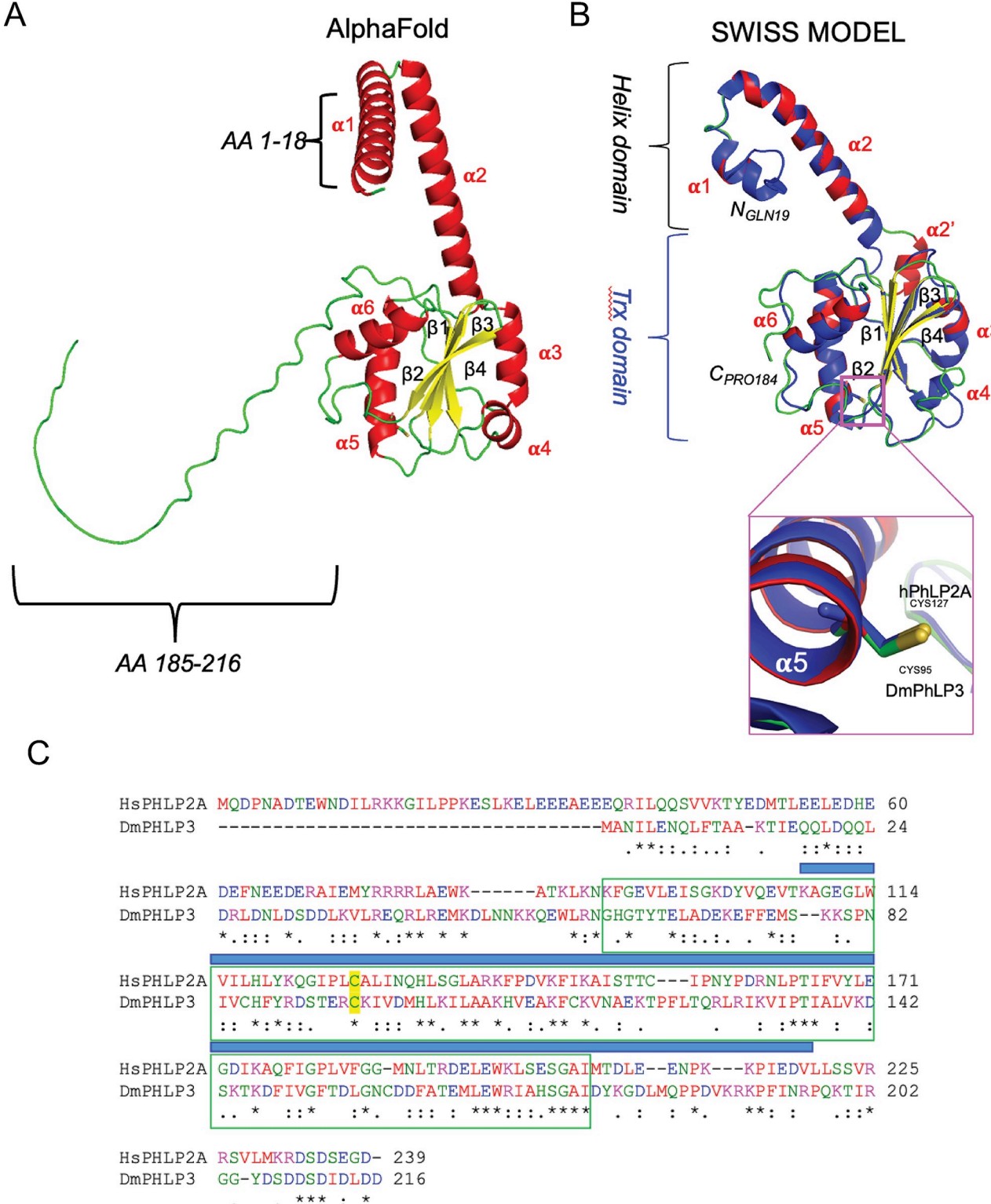

**Fig 4. AlphaFold and hypothetical model of *Drosophila melanogaster* PhLP3.** (A) AlphaFold prediction of the complete *D. melanogaster* PhLP3 structure (Q9VGV8; [29, 30]) showing the characteristic PhLP organization. Amino acid stretches missing at the N- and C-termini in the SWISS-MODEL are indicated. (B) Structure superposition of *D. melanogaster* PhLP3 (DmPhLP3) with human PhLP2A (hPhLP2A; PDB 7nvm: entity 11; [32]). Blue represents hPhLP2A, while red and green structures represent DmPhLP3. The green portions indicate unstructured segments, including loops and turns. The enhancement box shows the superposition for the conserved putative redox-active cysteines CYS95 (DmPhLP3) and

CYS127 (hPhLP2A). Structural alignment and rendering were prepared using PyMOL (Pymol.org, The PyMOL Molecular Graphics System, Version 1.2r3pre, Schrödinger, LLC). Alpha helix a2' in the SWISS-MODEL is part of helix a2 in the AlphaFold model. (C) Sequence alignment of DmPhLP3 with hPhLP2A.The pairwise sequence alignment was generated using Clustal Omega [15]. Amino acids are colored by polarity. The open green box represents the thioredoxin domain, and the blue bar above the sequence indicates the sequence portion modeled by SWISS-MODEL shown in B. * = identical AA;: = highly similar AA;. = similar AA. NCBI reference sequences: hPhLP2A NP_076970 [10], DmPhLP3 AAL28410.

(S2 Fig) [21–23]. To explore the expression of *PhLP3* in reproductive tissue, we performed *in situ* hybridization combined with immunostaining on dissected testes. Our data indicate that the gene is expressed in germ cells from the spermatogonia stage until the early spermatid stage but is no longer observed in late-stage spermatids in the distal end of the testis (Fig 6A–6C).

Given the expression of *PhLP3* in the germline until early spermatid stages, we next asked whether PhLP3 is required to produce mature sperm. We obtained a fly strain (*PhLP3^{EY13373}*) carrying a P-element insertion in the 5' UTR of *PhLP3*, located 78 bp upstream of the start codon (Fig 3) [38]. The location of the P-element in the 5' UTR of *PhLP3* was confirmed by inverse PCR and sequencing (S3 Fig). Homozygous *PhLP3^{EY13373/EY13373}* male flies and control male flies (*w^{1118}*) were mated to virgin female control flies (*w^{1118}*). Surprisingly, each *PhLP3* mutant cross failed to yield progeny over six independent crosses (Fig 7). In contrast, the control crosses yielded an average of 108 progeny over six independent crosses (Fig 7). Based on these results, it appears that PhLP3 mutant male flies are sterile, which suggests that PhLP3 is required for male fertility.

We hypothesized that the P-element insertion in the 5' UTR of *PhLP3* causes downregulation of *PhLP3* expression and links to the observed sterile phenotype. To test this, we performed real-time quantitative PCR (RT-qPCR) analysis on total RNA extracts from testes of *PhLP3* mutant and control flies. In three independent experiments, we show that *PhLP3* expression in mutant testes was downregulated by 94% when compared to control testes (Fig 6G). Similarly, fluorescent *in situ* hybridization of homozygous *PhLP3^{EY13373/EY13373}* mutant testes exhibited only weak fluorescence for the *PhLP3* probe, confirming our RT-qPCR results

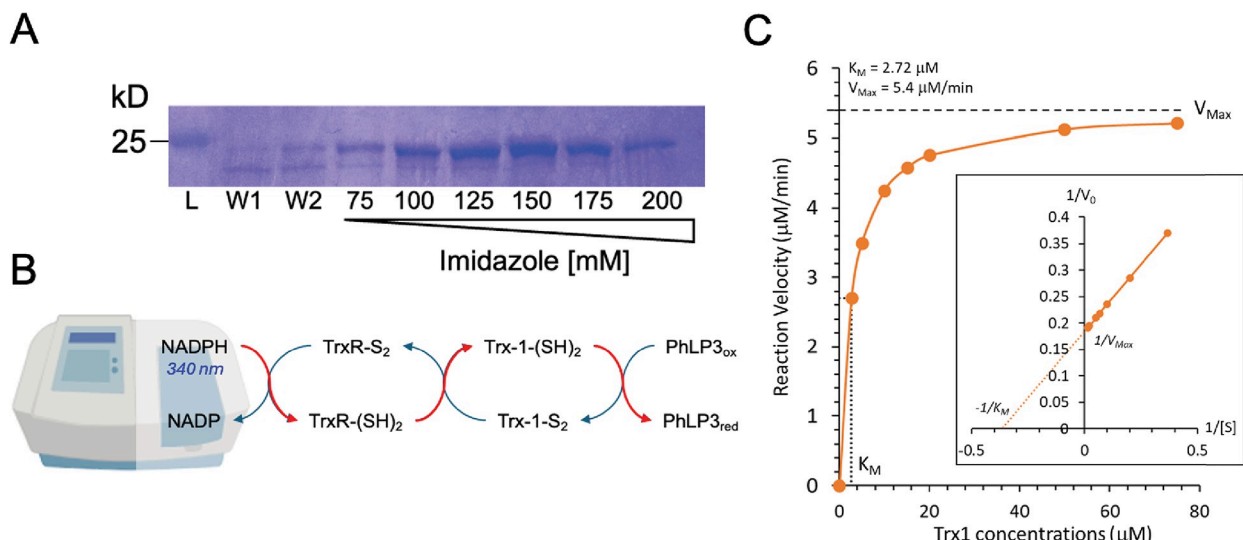

**Fig 5. Purification and enzyme activity of PhLP3.** (A) SDS gel following protein purification of recombinant His-tagged PhLP3. Washes (W) and elutions with increasing imidazole concentrations are indicated. (B) Principal setup of an *in vitro* thioredoxin reduction assay. The oxidation of NADPH to NADP is monitored at 340 nm. Red arrows indicate electron-flow from NADPH to Thioredoxin reductase (TrxR) to thioredoxin (Trx-1) and then to PhLP3, which will be reduced in the process. (C) Michaelis Menten diagram showing activity of PhLP3 with increasing thioredoxin concentrations. The indicated $K_m$ and $V_{max}$ values were determined using curve fitting. The values were verified by generating a Lineweaver Burk plot (inset).

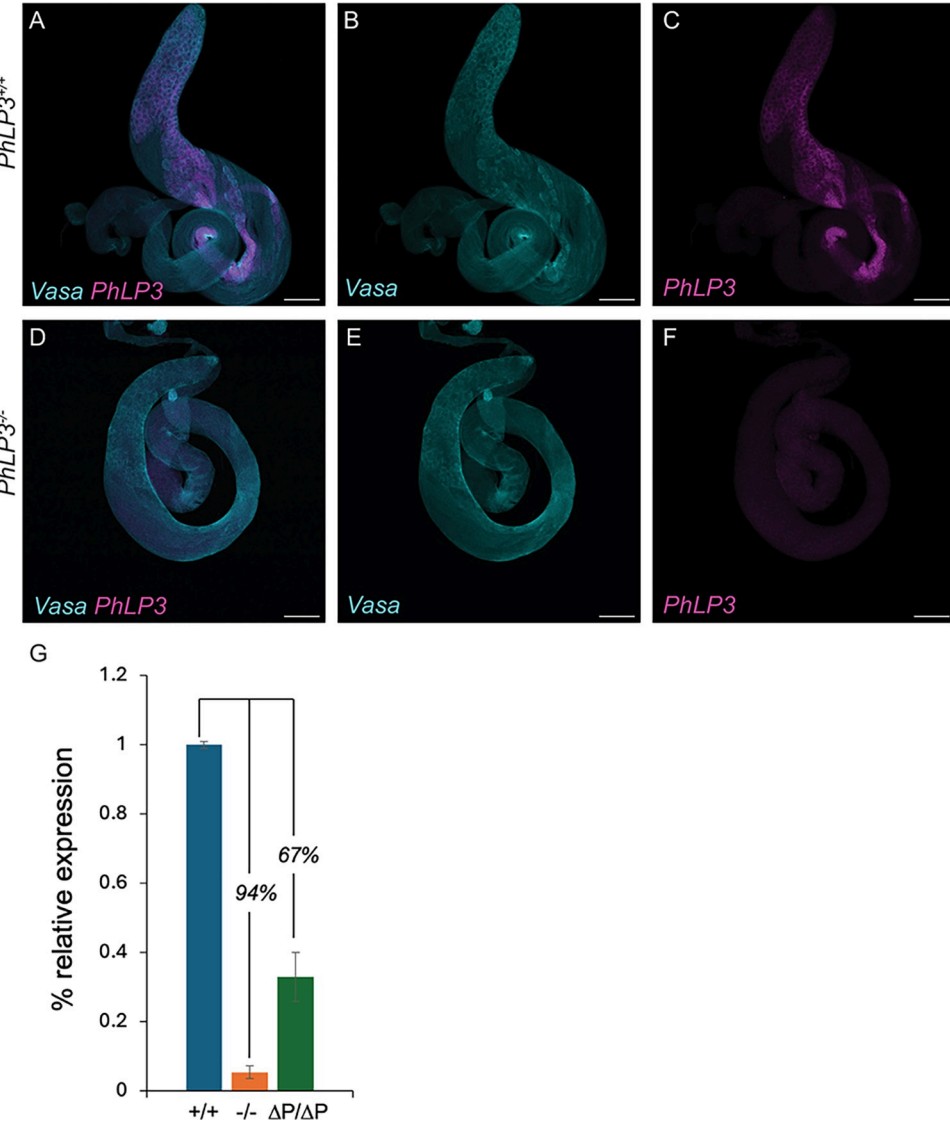

**Fig 6. Expression of *PhLP3* in the testes.** (A-F) Fluorescent *in situ* hybridization for *PhLP3*. *PhLP3* probe (magenta) and anti-Vasa to label germ cells (cyan). Scale bar is 100 μm. (A) *PhLP3*<sup>+/+</sup> testes exhibit expression in germ cells from spermatogonia through early spermatid stages (n = 59). (B) Germ cells immunostained with anti-Vasa (cyan). (C) *In situ* hybridization for *PhLP3* (magenta). (D) *PhLP3*<sup>-/-</sup> testes exhibit a loss of *PhLP3* expression throughout the testis (n = 36). (E) Germ cells immunostained with anti-Vasa (cyan). (F) *In situ* hybridization for *PhLP3* (magenta). (G) RT-qPCR shows downregulation of *PhLP3* expression in testes from males homozygous for the P-element insertion in the 5' UTR of *PhLP3* (-/-), compared to wild-type (+/+) levels. Excision of the P-element (ΔP/ΔP) partially rescues *PhLP3* expression.

and the specificity of our probe (Fig 6D–6F). Given the significant reduction in *PhLP3* expression in this mutant strain, we will refer to these mutants as *PhLP3*<sup>-/-</sup> for the remainder of the paper. These data strongly support our hypothesis that PhLP3 is required for male fertility in *D. melanogaster*.

## PhLP3 downregulation results in defects in sperm development

Based on the infertility observed in *PhLP3*<sup>-/-</sup> mutants, we determined whether mature sperm had been produced. We visually examined the testes and seminal vesicles of 1–3 day-old

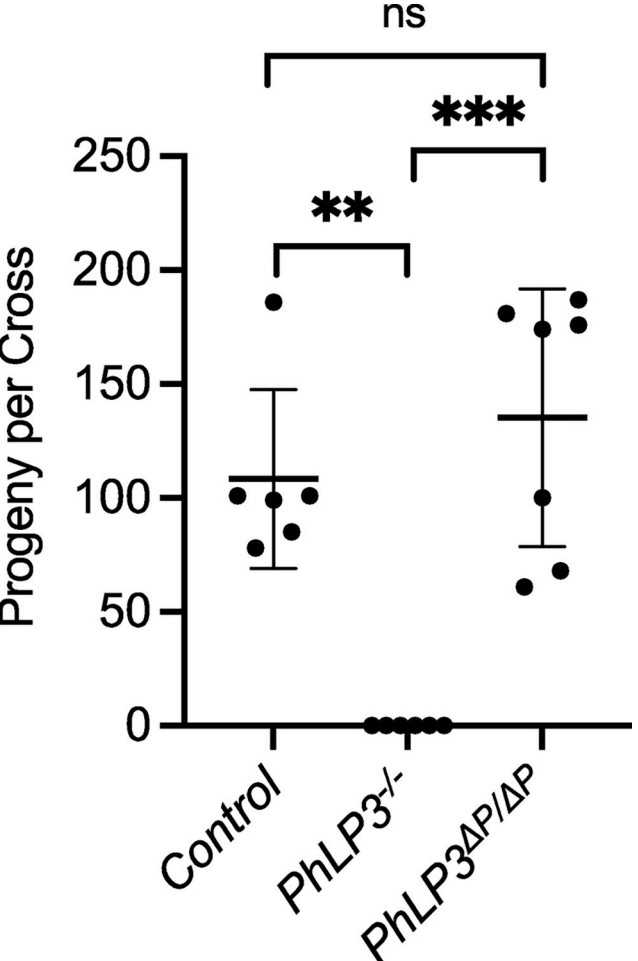

**Fig 7. Fertility tests.** Control males ($w^{1118}$), $PhLP3^{-/-}$ ($PhLP3^{EY13373/EY13373}$) males or P element excision males ($PhLP3^{\Delta P/\Delta P}$) were mated to $w^{1118}$ virgin females to assess fertility. The average number of progeny per cross was counted. Individual measurements for each cross are shown with the average and standard deviation indicated. *** indicates p<0.001 and ** indicates p<0.01 based on pairwise analysis by Welch's t-test; ns = not significant.

control and mutant male flies using DAPI staining and fluorescence microscopy (Figs 8 and 9). In the control testes, we observed the characteristic DAPI clusters of elongating sperm nuclei aligned in a syncytium near the distal end of the testes, giving way to a large seminal vesicle filled with mature sperm (Figs 8A–8E and 9A–9C). In contrast, $PhLP3^{-/-}$ mutants lacked the aligned DAPI-stained nuclear clusters (Fig 8F–8H). Instead, the nuclei appeared scattered and failed to progress from the canoe stage to the needle-like nuclei, which would indicate maturing sperm (Fig 9D–9F). Furthermore, compared to the controls, the size of the seminal vesicles in the mutant testes was diminished by 32%, and seminal vesicles lacked mature sperm (Fig 8I, 8J and 8P).

Considering the absence of mature sperm in the seminal vesicles, we hypothesized that spermatids in $PhLP3^{-/-}$ mutants are unable to initiate the process of individualization. Individualization of spermatids by actin-based sperm individualization complexes is an essential process that functions to remove excess organelles and cytoplasm and to encapsulate each spermatid with its own plasma membrane. To determine if individualization complexes could form, we used phalloidin staining to detect the presence of the characteristic actin cones.

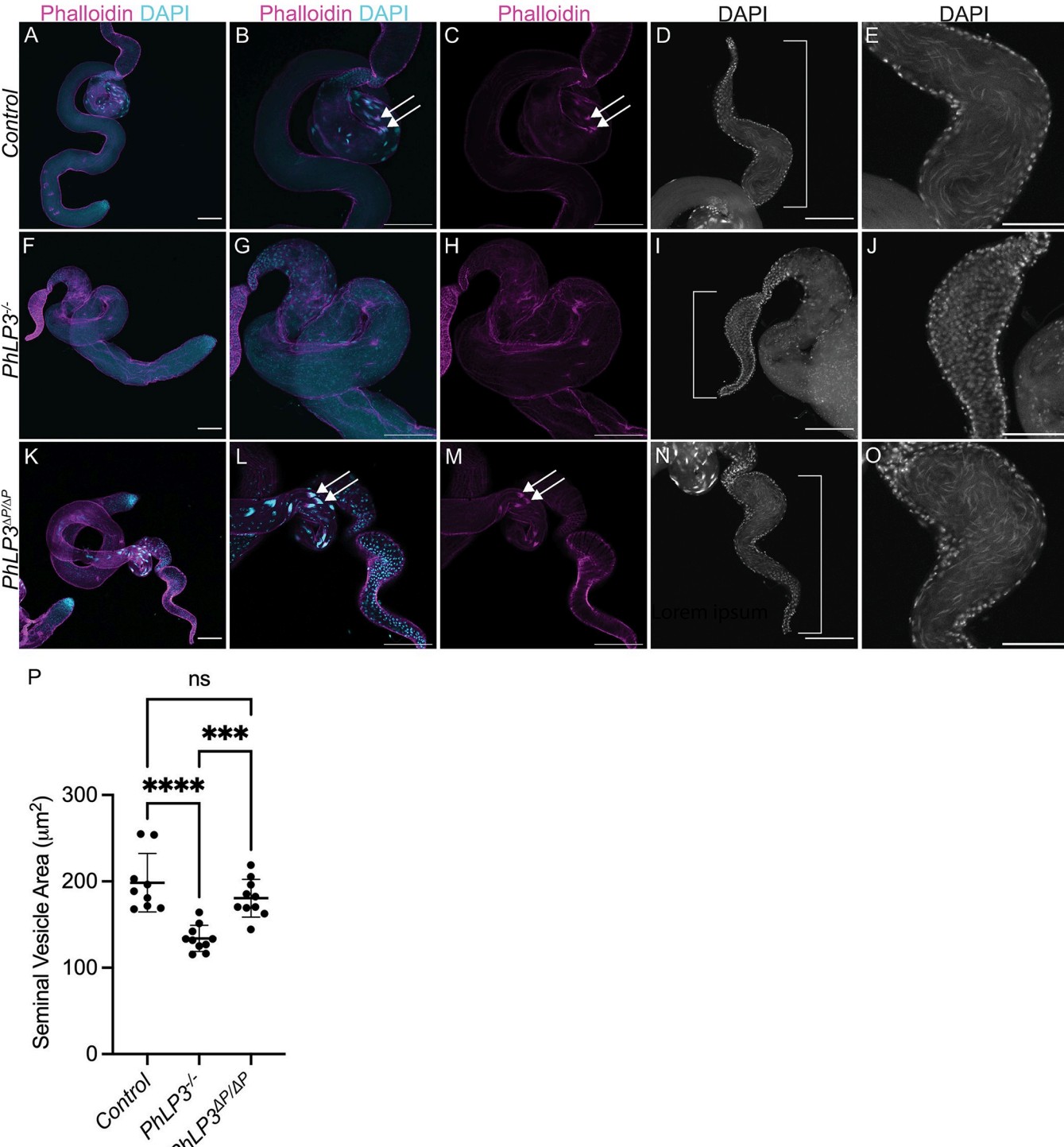

**Fig 8. Mutation of *PhLP3* causes arrest of spermiogenesis.** (A-C) Control *PhLP3*$^{+/+}$ testes have DAPI clusters and actin-based individualization complexes (n = 40). Nuclei stained with DAPI (cyan) and actin-based individualization complexes stained with phalloidin (magenta). (B-C) Higher magnification view of testis in (A). Arrows indicate individualization complexes. (D-E) Control *PhLP3*$^{+/+}$ seminal vesicle. (D) Seminal vesicle is large with mature sperm (n = 21). Nuclei stained with DAPI. (E) Zoomed in view of the seminal vesicle from (D) filled with mature sperm. (F-H) *PhLP3*$^{-/-}$ testes have dispersed DAPI clusters, less elongated nuclei, and no actin-based individualization complexes (n = 38). Nuclei stained with DAPI (cyan) and actin-based individualization complexes stained with phalloidin (magenta). (G-H) Higher magnification view of testis in (F). (I-J) *PhLP3*$^{-/-}$ seminal vesicle. (I) Seminal vesicle is smaller than control and lacks sperm (n = 15). Nuclei stained with DAPI. (J) Zoomed in view of the seminal vesicle from (I) lacking mature sperm. (K-M) Excision of the P element in *PhLP3* (*PhLP3*$^{\Delta P/\Delta P}$) rescues the mutant phenotype. Testis contain DAPI clusters and individualization complexes (n = 23). Nuclei stained with DAPI (cyan)

and actin-based individualization complexes stained with phalloidin (magenta). (L-M) Higher magnification view of testis in (K). Arrows indicate DAPI clusters with individualization complexes. (N-O) *PhLP3*$^{\Delta P/\Delta P}$ seminal vesicle. (N) Seminal vesicle contains mature sperm (n = 19). Nuclei stained with DAPI. (O) Zoomed in view of the seminal vesicle from (N) filled with mature sperm. (P) Cross section area of the seminal vesicle (n = 10 for each genotype). Individual measurements are shown with the average and standard deviation indicated. **** indicates p<0.0001 and *** indicates p<0.001 based on ordinary one-way ANOVA; ns = not significant. Scale bars are 100 μM in A-D, F-I, and K-N. Scale bar is 50 μM in E, J, and O.

While control testes exhibited clusters of actin cones, typical of individualization complexes in the distal testes, *PhLP3*$^{-/-}$ mutants completely lacked actin cones, suggesting that spermatids failed to assemble individualization complexes (Figs 8A–8C, 8F–8H, 9B and 9E). This data suggests that *PhLP3*$^{-/-}$ mutant sperm development is arrested during the canoe stage of spermiogenesis, failing to reach the needle-like stage and progress through individualization. As a result, seminal vesicles are devoid of mature sperm and smaller in size.

## Excision of P-element in *PhLP3* restores male fertility

To confirm the specificity of the mutant phenotype observed in the P-element insertion in *PhLP3*, we excised the P-element from the *PhLP3* gene to reconstitute the 5' UTR, restore PhLP3 expression, and rescue male fly fertility. We confirmed the successful and precise excision of the P-element by sequencing and refer to the resulting strain as *PhLP3*$^{\Delta P/\Delta P}$ (S3 Fig). We first tested the fertility of *PhLP3*$^{\Delta P/\Delta P}$ male flies by mating them to control virgin females. An average of 135 *PhLP3*$^{\Delta P/\Delta P}$ progeny were produced over seven independent fertility crosses, compared to an average of 108 progeny for control crosses (Fig 7). This data confirms that the

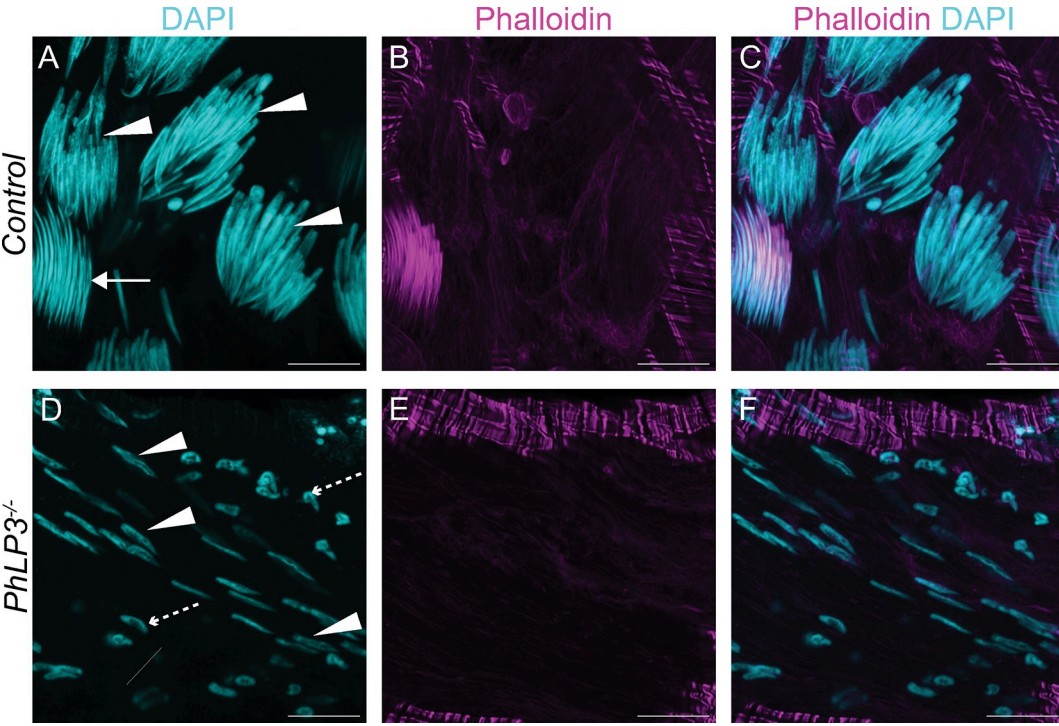

**Fig 9. Morphology of spermatids in *PhLP3* mutants.** (A-C) Control *PhLP3*$^{+/+}$ testes have clusters of spermatid nuclei undergoing multiple stages of spermiogenesis, as indicated by DAPI (cyan). Individualizing spermatids are stained with phalloidin (magenta). (n = 40). (B) *PhLP3*$^{-/-}$ testes have dispersed spermatid nuclei at various stages of elongation but lack needle-like nuclei, as indicated by DAPI staining (cyan). Individualization complexes are not observed by phalloidin staining (magenta) (n = 37). (A, D) Arrowheads label canoe stage nuclei. Dashed arrows label leaf stage nuclei. Solid arrows label needle-like nuclei. Scale bars are 10 μM. Images are maximum intensity Z projections of whole-mount testes.

P-element excision restored fertility to control levels and supports the hypothesis that PhLP3 is essential for male fertility. To determine whether gene expression was also restored, we performed RT-qPCR on total RNA from dissected *PhLP3*$^{ΔP/ΔP}$ and control testes. Remarkably, our results from five independent experiments show that *PhLP3* expression in the *PhLP3*$^{ΔP/ΔP}$ was restored to only 33% of control expression levels, suggesting there may be a relatively low threshold requirement for *PhLP3* expression and function in sperm development (Fig 6G). This was surprising, given that fertility was completely restored. Furthermore, visual analysis of *PhLP3*$^{ΔP/ΔP}$ testes revealed that sperm production was reestablished to levels comparable to control testes (Fig 8K–8O). DAPI-stained, aligned clusters of elongating spermatid nuclei in the *PhLP3*$^{ΔP/ΔP}$ testes are evident, as are the actin cones of the individualization complexes (Fig 8K–8M). Seminal vesicles were filled with mature sperm, and the seminal vesicle area was 91.4% of the size of the control testes (Fig 8N–8P). Sperm maturation appeared to proceed normally in *PhLP3*$^{ΔP/ΔP}$ testes despite only partially restored *PhLP3* gene expression (Fig 8K–8O). Our results confirmed that the P-element insertion in the 5' UTR of the *PhLP3* gene prevents effective gene transcription. Despite fully restored fertility, the excision of the P-element only restores transcription activity levels to 33%, introducing the possibility that other factors may be involved in the observed phenotype.

To affirm that the downregulation of *PhLP3* specifically causes the male sterile phenotype, we utilized RNA interference (RNAi) as an alternative approach to reduce *PhLP3* levels under the control of the Gal4/UAS system [39]. We used *bam*-Gal4 to drive the expression of a UAS-*PhLP3*-RNAi in the germline [40, 41]. The efficacy of this RNAi approach was tested by RT-qPCR and showed a 77% knockdown (23% of control) in *PhLP3* expression in *bam*-Gal4>*PhLP3*-RNAi testes compared to testes of the *bam*-Gal4 control (Fig 10A). We tested the fertility of these flies by mating *bam*-Gal4 control males or *bam*-Gal4>*PhLP3*-RNAi males to control virgin females. We scored the number of progeny and confirmed that the fertility was significantly reduced in *PhLP3*-RNAi flies, with an average of 3 progeny eclosing per cross. Control crosses yielded an average of 135 progeny per cross, a similar number of progeny to control crosses described earlier (Fig 10B). Next, we examined the effect of *PhLP3* RNAi knockdown on spermatogenesis by DAPI and phalloidin staining and confocal microscopy. The *bam*-Gal4 control testes exhibited spermatids that clustered, elongated, and individualized (Fig 10C–10F). In contrast, while *bam*-Gal4>*PhLP3*-RNAi testes possessed spermatids, nuclei failed to elongate fully and appeared scattered throughout the syncytium they developed within, similar to our observation of the P-element containing *PhLP*$^{-/-}$ flies (Fig 10I–10L). Using phalloidin staining, we observed clustered and aligned actin cones in individualization complexes in *bam*-Gal4 control testes (Fig 10D and 10E), but actin cones appeared scattered in *PhLP3*-RNAi testes (Fig 10J and 10K). Consistent with the reduced fertility of *PhLP3*-RNAi flies compared to controls, seminal vesicles lack the mature sperm with needle-like nuclei observed in controls, demonstrating a failure to complete sperm maturation (Fig 10G, 10H, 10M and 10N). Our results provide strong evidence for the specificity of the phenotype to the *PhLP3*$^{-/-}$ mutant and demonstrate an essential requirement for PhLP3 in *D. melanogaster* spermiogenesis and male fertility.

## Materials and methods

### Fly strains

The following fly strains were utilized in these studies: *w*$^{1118}$; *Dr/TM6B, Tb, Hu, Deformed (Dfd)-Yellow Fluorescent Protein (YFP)* [42]; *PhLP3*$^{EY13373/+}$/*TM3, Sb* or *PhLP3*$^{EY13373/EY13373}$ (Bloomington *Drosophila* Stock Center (BDSC #21414; [38]); *w\*; Dr*$^{1}$/*TMS, P{ry [+7.2] Δ2–3} 99B* (BDSC #1610); *bam*-Gal4 (BDSC #80579; [40]); and *UAS-PhLP3-RNAi* (Vienna

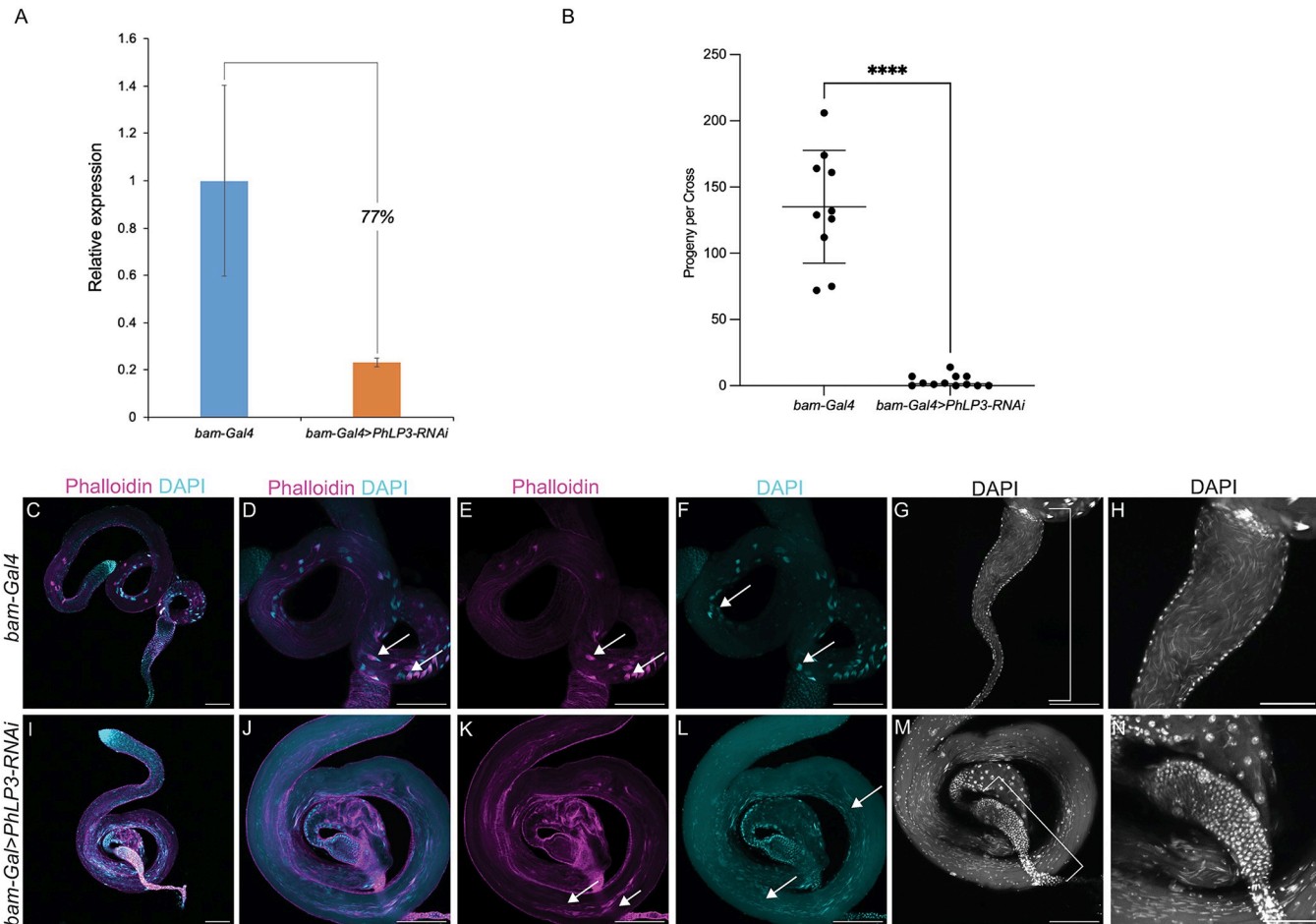

**Fig 10. Effects of *PhLP3* RNAi on mature sperm production.** (A) RT-qPCR demonstrates a 77% reduction in *PhLP3* expression by RNA-interference (RNAi) in comparison to that of control (*bam-Gal4*) samples. (B) Fertility cross data showing the number of progeny resulting from four virgin female $w^{1118}$ flies crossed with three male *bam-Gal4* control flies or three male *bam-Gal4>PhLP3-RNAi* flies. Individual measurements for each cross are shown with the average and standard deviation indicated. **** indicates p<0.0001 based on Welch's t-test. (C-F) *bam-Gal4* control testis (n = 47). Whole testis stained with DAPI (cyan) stain to visualize nuclei and phalloidin (magenta) to visualize actin-based individualization complexes. (D-F) Higher magnification image of testis from (C). (D) Arrows indicate colocalized DAPI clusters with actin-based individualization complexes (magenta). (E) Arrows indicate actin-based individualization complexes. (F) Arrows indicate clusters of aligned, elongating spermatid nuclei stained with DAPI. (G) *bam-Gal4* control seminal vesicle filled with mature sperm indicated by bracket (n = 24). (H) Zoomed in view of seminal vesicle from (G) filled with sperm (G). (I-L) *bam-Gal4>PhLP3-RNAi* testis (n = 29). Whole testis stained with DAPI (cyan) stain to visualize nuclei and phalloidin (magenta) to visualize actin-based individualization complexes. (J-L) Higher magnification image of testis from (I). (J) There is little colocalization of DAPI with actin-based individualization complexes. (K) Arrows indicate scattered actin cones. (L) Arrows indicate scattered, elongating spermatid nuclei that fail to cluster as compared to controls. (M) *bam-Gal4>PhLP3-RNAi* seminal vesicle indicated by bracket is reduced in size and lacks mature sperm (n = 19). (N) Zoomed in view of the seminal vesicle from (M) lacking mature sperm. Scale bar is 100 μM in C-G and I-M. Scale bar is 50 μM in H and N.

*Drosophila* Resource Center #100784; [41]). Flies were raised at 25°C on standard Bloomington *Drosophila* cornmeal medium.

The strain in which the P-element was precisely excised from the *PhLP3* gene was generated as follows: virgin females of the genotype *PhLP3$^{EY13373}$/TM3, Sb* were mated to males of the genotype *w\*; Dr$^1$/TMS, P{ry [+7.2] Δ2–3}99B*. Male progeny of the genotype *PhLP3 $^{EY13373}$/ TMS, P{ry [+7.2] Δ2–3}99B* were collected and mated to *Dr/TM6B, Tb, Hu, Dfd-YFP* virgin females. Single white eyed progeny of the potential genotype *PhLP3$^{ΔP}$/ TM6B, Tb, Hu, Dfd-YFP* were selected and mated to *Dr/ TM6B, Tb, Hu, Dfd-YFP* flies of the opposite sex to establish stocks. Stocks were screened by PCR and sequencing as described below.

## PCR and sequencing

To isolate genomic DNA for inverse PCR (iPCR) of the P element insertion, fifteen 1–3 day old *PhLP3$^{EY1337/ EY13373}$* or control *(w$^{1118}$)* flies were collected and homogenized in 400 μl Buffer A (100mM Tris pH 7.5, 100 mM EDTA, 100 mM NaCl, 0.5% SDS), followed by a 30 min incubation at 65˚C. 800 μl LiCl/KAc (2.28 M LiCl, 1.43 M KAc) was added and samples were incubated on ice for 10 min. Samples were spun for 15 min at 12,000 rpm at room temp. 1 mL of supernatant was transferred to a new tube and 600 μl of isopropanol was added, and tube was inverted to mix. Samples were spun for 20 min at 12,000 rpm at room temp. Supernatant was discarded and the samples were spun down quickly to remove additional supernatant. The pellet was washed with 500 μl cold 70% ethanol and spun down for 10 min at 12,000 rpm at room temp. Supernatant was discarded and the samples were spun down quickly to remove additional supernatant. Pellet was air dried and resuspended in 75 μl 1x TE (10mM Tris pH 7.5, 1 mM EDTA). Genomic DNA was then digested with HpaII at 37˚C for 2.5 hours, followed by a 20 min heat inactivation at 65˚C. NEB T4 DNA Ligase was then utilized in a ligation reaction to form a plasmid library from our genomic digest, and an inverse PCR was performed to amplify the 5' and end of the P{EPgy2} P-element insertion, using the Plac1 and Pwht1 primers (S1 Table) [38]. Products were sequenced, using the 5.SUP.seq1 primer by Genewiz/Azenta.

For sequencing across the genomic region of control *(w$^{1118}$)* and P element excision flies, single flies were collected and homogenized using a pipette tip filled with 50 μl of Squishing Buffer (10 mM Tris-Cl pH 8.2, 1mM EDTA, 25 mM NaCl and 200 μg/mL Proteinase K) [43]. Samples were incubated for 20 min at 37˚C. Proteinase K was heat inactivated for 2 min at 95˚C and DNA was stored at 4˚C. The genomic region around the site of the P element insertion was amplified by PCR using the CG4511-Fwd and CG4511-Rev primers and sequenced using the same primers by Genewiz/Azenta.

## Fertility crosses

Four *w$^{1118}$* virgin females were crossed to three male flies of genotype *w$^{1118}$*, *PhLP3$^{-/-}$*, homozygous P-element excision *(PhLP3$^{ΔP/ΔP}$)*, *bam-Gal4, or bamGal4>PhLP3-RNAi*. Flies were mated for 7 days. Parental flies were emptied from the bottle on day 7, and progeny were collected over a period of five days (days 10–15) and scored. Statistical analysis was carried out with GraphPad Prism.

## Immunohistochemistry

Testes from 1–3 day old males were processed as described previously in the presence of 5% NGS [44]. Following antibody staining, samples were incubated with Phalloidin conjugated to Alexafluor 555 diluted 1:40 in 1x PBS for 10 minutes, washed for 5 minutes in 1x PBS, and mounted in DAPI Fluoromount-G (Southern Biotech). Images were acquired on a Zeiss LSM880 confocal microscope and processed using Fiji [45]. The cross-section area of seminal vesicles was measured in Fiji and statistical analysis of data was carried out in GraphPad Prism.

## RT-qPCR

Testes from 1–3 day old *w$^{1118}$*, *PhLP3$^{-/-}$*, or *PhLP3$^{ΔP/ΔP}$* males were dissected in RNase-free (DEPC-treated) PBS and incubated in 1 mL RNAzol$^{TM}$ (Molecular Research Center) at 4˚C. Total RNA was extracted following the manufacturer's instructions. Isolated RNA was treated with DNAse I (Ambion) and subsequently quantified using a Qubit RNA Assay kit (Invitrogen). RNA samples were immediately used for cDNA synthesis employing a high-capacity

RNA-to-cDNA kit (Applied Biosystems) with random hexamer primers. RT-qPCR was performed on a StepOnePlus machine (Applied Biosystems) using the PowerUp SYBR Green Master Mix (Applied Biosystems). Each sample was run in triplicate and yielded highly comparable Ct values (cycle threshold). No primer dimers were detected, and amplicons exhibited optimal efficiencies. Expression data was analyzed using the StepOne Software v2.2 (Applied Biosystems) and normalized against the expression of the endogenous *glyceraldehyde 3-phosphate dehydrogenase* (*gapdh*) enzyme, which is an established internal standard for expression analysis in *D. melanogaster* [46]. *Ribosomal protein L32* (*RpL32*) was used as an additional internal positive control [47]. Primers are listed in S1 Table.

## Cloning, expression, and purification of recombinant PhLP3

Gene-specific primers (S1 Table) for *PhLP3* (FlyBase *CG4511*, NCBI NP_001163585) were generated according to sequence information on FlyBase and NCBI [20]. PCRs were performed using the following conditions: 35 cycles of 95°C for 30 s, 54°C for 1 min, and 72°C for 45 s. Following sequence verification, the *PhLP3* CDS was cloned into the bacterial pQE9 expression vector (Qiagen), which introduced a 6x HIS tag at the N-terminus of the recombinant protein. The *PhLP3* expression plasmid was transformed into *E. coli* M15 expression cells (Qiagen). Protein expression was induced (1 mM IPTG), and bacteria were harvested after a 5-hour incubation period at 37°C. Recombinant proteins were purified under denaturing conditions (8 M urea) using Ni-NTA resin (Thermo Scientific). The purity of the recombinant proteins was assessed via SDS-PAGE. Protein concentrations were determined using the Qubit fluorometer (Invitrogen) and Bradford assay.

## Enzyme assays

All enzymatic assays were carried out in 1 ml volume at 25°C using a Genesys6 UV-Vis spectrophotometer (Thermo Fisher). *P. berghei* Thioredoxin reductase (PbTrxR, PbANKA_0824700, [36]) and *D. melanogaster* thioredoxin-1 (DmTrx-1, CG4193, [37]). were produced as described previously [4, 35]. The coupled enzymatic assays were conducted as previously described [4]. Briefly, the oxidation of NADPH was followed by an absorption decrease at 340 nm. All assays were performed at RT in assay buffer containing 100 mM $KH_2PO_4$, 2 mM EDTA, pH 7.4, 200 μM NADPH ($\varepsilon340$ nm = 6.22 $mM^{-1}cm^{-1}$), and 20 μM *D. melanogaster* Trx-1. Each initial reaction was started with PbTrxR [4, 35], and the decrease in absorption at 340 nm was monitored during the linear phase. Once the reaction was complete, purified recombinant *D. melanogaster* PhLP3 was added, and the initial velocities and kinetic values for each reaction were determined using the VISIONlite (Thermo Fisher) software. Enzyme kinetics were calculated using global curve fit in the Enzyme Kinetics Module of Sigma Plot 12.0.

## Comparative modeling

The hypothetical models of *D. melanogaster* PhLP1, 2, and 3 were generated using SWISS-MODEL [34]. The crystal structure of the human PhLP2A (PDB ID: 7nvm [17]) was selected as the template from the RCSB Protein Data Bank (http://www.rcsb.org/). The models were visualized using the PyMOL (Pymol.org, The PyMOL Molecular Graphics System, Version 1.2r3pre, Schrödinger, LLC).

## *In situ* hybridization

Testes from $w^{1118}$(*PhLP3*$^{+/+}$) and *PhLP3*$^{-/-}$ males were dissected and processed for fluorescent *in situ* hybridization combined with immunostaining as described previously [48]. Probes

were prepared by Molecular Instruments and were conjugated to AlexaFluor 488. Samples were immunostained with rat anti-Vasa (1:50; Developmental Studies Hybridoma Bank [49]) and goat anti-rat AlexaFluor 555 (1:500) to label germ cells. Samples were imaged on a Zeiss LSM880 confocal microscope, and images were processed using Fiji [45].

## Discussion

In this study, we identified the *D. melanogaster* homolog of the PhLP3 group of Phosducin-like proteins as the protein encoded by the *CG4511* gene. Phosducin-like proteins (PhLPs) are members of the thioredoxin (Trx) superfamily and have been identified in eukaryotes from yeast to humans [2, 7]. They show surprisingly high sequence conservation across species and have been organized into three subgroups based on sequence similarities: PhLP1, PhLP2, and PhLP3 [7]. Three phosducin-like genes have been identified in the genome of *D. melanogaster*, namely *CG4511*, *CG7650*, and *CG18593* (S1 Fig). BLAST analysis confirmed that the protein encoded by *CG4511* readily aligns with members of the PhLP3 group, some of which have been implicated in the regulation of the cytoskeleton in *S. cerevisiae*, *C. elegans*, *A. thaliana*, and human cell lines [2, 6, 7, 12, 16]. Of the three genes, *CG4511* (*PhLP3*) exhibits high transcription levels in the testes and ovaries (S2 Fig) [21, 22] and was thus chosen as our primary target. The CG7650 protein aligns most closely with PhLP1 group members, some of which are involved in trimeric G-protein folding and regulation [5, 8]. A recent CryoEM study of the human CCT complex with PhLP1 bound gave a high-resolution insight that suggests that human PhLP1 assists in the folding, stabilizing, and releasing of the G-protein beta subunit from the cavity of the CCT complex, an essential step in forming the obligate G-protein beta-gamma dimer [50]. The CG18593 protein, annotated as Viral Inhibitor of Apoptosis-associated Factor (Viaf) in FlyBase, favorably aligns with the PhLP2 group (S1 Fig). In yeast, PhLP2 (Plp2) assists in the folding and assembly of actin and tubulin subunits, respectively [10]. Recent CryoEM studies show human PhLP2A (PDCL2)-actin and PhLP2A-tubulin complexes in the cavity of the CCT chaperonin complex, further supporting the hypothesis that PhLP family members are involved in the folding and regulation of cytoskeletal proteins [17, 32]. Notably, two recent reports implicate the mouse PhLP2 (PDCL2) in the spermiogenesis of adult male mice [51, 52]. The most recent data on PhLPs from all groups indicate interactions with the cellular protein folding machinery, specifically CCT and Heat shock proteins [53]. The precise mechanism of how PhLPs assist in this mechanism is not yet known.

*D. melanogaster* PhLPs share the same structural organization as other PhLPs, consisting of a central thioredoxin domain flanked by an N-terminal alpha helix domain and a short, variable, and mostly unstructured C-terminal tail [1]. The thioredoxin domain shows the highest degree of sequence and structure conservation between different organisms, with *D. melanogaster* PhLP3 sharing 43% sequence identity with the protozoan organism *P. berghei* and even 65% identity with the human homolog TXNDC9. While the N-terminal sequences may differ significantly between species, they all form or are predicted to form two to three alpha-helices, depending on the PhLP group. The PhLP1 group with a longer N-terminus contains three alpha-helices, while members of the PhLP2 and PhLP3 group seem only to form two helices. Stirling et al. [6] demonstrated that the alpha-helix domain, as well as the C-terminal tail, are required for PhLP3 to interact with the CCT complex. As the alpha-helix structure is being maintained, this may reflect species-specific sequence variations, while the highly conserved thioredoxin domain may possess an equally conserved functional mechanism involved in target protein folding or stabilization within the CCT complex.

Many thioredoxins and thioredoxin-like proteins exhibit redox activity, often facilitated by a pair of cysteines in a characteristic CXXC active site motif [33]. The lack of this motif in

PhLPs prompted the dismissal of potential redox activity for these proteins [7]. Yet, our group recently found that *P. berghei* PhLP3, as well as the human TXNDC9, exhibit redox activity with the thioredoxin system and that this redox activity is facilitated by a highly conserved cysteine residue exposed at the top of alpha-helix 2 in the thioredoxin domain (Figs 1 and 5) [4]. This cysteine is conserved in *D. melanogaster* PhLP3 (CYS95), and it was therefore not surprising to find that the recombinant PhLP3 described in this paper also exhibits redox activity with thioredoxin. Our kinetics data compares very well with that of previously characterized PhLP3s [4]. This is significant as the thioredoxin system plays an important role in the cellular redox homeostasis of *D. melanogaster* [35, 54]. Interestingly, two germline-specific thioredoxins have recently been reported in *D. melanogaster* Deadhead (Dhd) and Thioredoxin T (TrxT) [55]. Both are hypothesized to play important roles in redox homeostasis in the germline of *D. melanogaster*. Notably, the thioredoxin used in this report to demonstrate electron transfer from the thioredoxin system to PhLP3 is Dhd (Trx-1, CG4193; [37]). Considering the high abundance of Dhd in ovaries and TrxT in the testes, it is possible that PhLP3 is a specific redox target for these proteins *in vivo*. Whether TrxT can also transfer electrons to PhLP3 remains to be determined.

The thioredoxin system is a central electron distribution system. Some of its target proteins are thioredoxin-like proteins with strong antioxidant activity [56]. We previously demonstrated that *P. berghei* PhLP3 and its homolog human TXNDC9 exhibit antioxidant activity that is orders of magnitude weaker than that of antioxidant effectors, such as thioredoxin-dependent peroxiredoxin (TPx1) [4]. We concluded that PhLP3 is unlikely to act as an antioxidant scavenger in the cell. In the fruit fly effective antioxidant systems, such as catalase and superoxide dismutase, cooperate in the fly to protect from reactive oxygen species [57]. The hypothesis for a different function of PhLP3 is corroborated by the recent structural insights into the interactions of PhLP3 and PhLP2A with the CCT complex and its potential role in the folding of actin and tubulin [6, 17]. It is feasible to hypothesize that the redox activity of PhLP3 plays a role in the mechanism that facilitates folding and/or release of target proteins in and from the CCT complex. Members of the thioredoxin superfamily are known to be involved in redox-assisted protein folding [33]. For example, protein disulfide isomerases utilize their redox activity to assist protein folding in the endoplasmic reticulum [58]. Supporting our hypothesis is the fact that the redox-active cysteine seems to be conserved in all PhLPs, notably between PhLP3 and human PhLP2A. We are currently investigating the role of the redox active cysteine in the cellular function and possible enzymatic mechanism of PhLP3.

At the cellular level, the primary defects observed in *PhLP3* mutants occur in the post-meiotic maturation stages, as spermatid nuclei elongate and spermatids undergo individualization. These processes are dependent on tubulin and actin. As PhLP3 proteins in other species have been demonstrated to regulate CCT function [6], mutation of *PhLP3* could affect the folding of tubulin and actin subunits, leading to sperm maturation defects. Mutations in α- and β-tubulin subunits have been demonstrated to exhibit a variety of germ cell defects, ranging from defects in meiosis, nuclear elongation and alignment, and axoneme assembly [59–65]. While the function of only one α-tubulin, α1-tubulin (encoded by *αTub84B*), has been implicated in the post-mitotic germ cells, two β-tubulins have been identified to function post-mitotically in sperm development [65, 66]. β1-tubulin (encoded by *βTub56D*) is present in testis through the spermatocyte stage when it is downregulated, with only low levels being detected in the spermatids heads, while β2-tubulin (encoded by *βTub85D*) begins to be detected in spermatocyte stages and through the remaining stages of sperm development [67]. Mutant alleles, resulting in an imbalance in α and β tubulin levels, lead to defects in spermatogenesis. For some mutant β-tubulin alleles, the resulting defects are similar to those we have observed in *PhLP3* mutants, including defects in nuclear shaping and alignment [66].

Nuclear shaping and alignment appear to depend on the establishment of the microtubule organizing center post-meiotically during the round spermatid stage. In the testis, the centriole adjunct serves at the main microtubule organizing center, and the establishment of the centriole adjunct depends on the proper docking of the tubulin-based basal body at the nuclear envelope. The centriole adjunct nucleates the perinuclear microtubules of the dense complex (analogous to the nuclear manchette in mammals) and the tail microtubules (reviewed in [68]). In mutants for the γ-Tubulin Ring Complex subunits, *grip75* and *grip128*, the basal body fails to dock at the nuclear envelope, and the nuclei appear misshapen and are scattered throughout the syncytium, resulting in sterility, similar to the defects observed in our *PhLP3* mutants [69]. Mutation of other genes encoding proteins believed to promote attachment of the basal body to the nuclear envelope, including *sperm-associated antigen 4 (spag4)* and y*uri gagarin* (y*uri*), exhibit similar nuclear scattering defects, failed individualization, and male sterility [70, 71]. Thus, failed docking of the tubulin-based basal body appears to result in nuclear shaping and alignment defects. Others have suggested these defects can also arise later due a failure to form and/or organize the perinuclear microtubules of the dense complex [63, 72].

The dense complex begins to form in the round spermatids in the region where the nucleus is flattened and fenestrated near the basal body [25]. This region invaginates to form a concavity filled with microtubules during nuclear elongation, while a single layer of microtubules is observed around the convex region of the nucleus [25]. These microtubules are believed to provide support for the nucleus during elongation. More recent work suggests the presence of a second microtubule organizing center at the tip of the nucleus from which some perinuclear microtubules of the dense complex may emanate [73]. Interestingly, *Sas-4* mutants lack basal bodies but still form dense complexes, suggesting this second microtubule organizing center could serve as a significant site of perinuclear microtubule nucleation [73]. Mutation of *PhLP3* could lead to defects in basal body docking and/or formation of the microtubule-rich dense complex, which appear to be required for nuclear elongation and alignment.

As PhLP3 proteins have also been demonstrated to regulate CCT-mediated folding of actin [6], sperm maturation defects in *PhLP3* mutants could also result from reduced actin levels. Mutations in *yuri* result in a loss of actin accumulation in the dense complex, suggesting that defects in actin folding could affect dense complex quality and nuclear elongation [70]. Live imaging of syncytial cysts reveals that actin is first observed around the nucleus during the canoe stages and later accumulates at the base of the nucleus before moving down the elongated spermatids for the process of individualization [26, 27]. If PhLP3 regulates folding of actin subunits as part of the CCT, a lack of available subunits would result in a failure to incorporate actin into dense complexes, which may contribute to the defects in nuclear elongation and alignment. In *PhLP3* mutants, we do not observe the formation of actin-based individualization complexes, which could arise from reduced actin levels or a failure to reach the individualization stage of spermiogenesis. Interestingly, mutants for *mulet*, which encodes the Tubulin-binding cofactor E-like (TBCEL), exhibit scattering of nuclei and defects in individualization; however, individualization complexes still form in these mutants and persist around scattered nuclei [74–76]. TBCEL appears to be required for microtubule disassembly to promote individualization. Other individualization mutants described by Fabrizio have disrupted individualization complexes or complexes that fail to migrate to the tail end of the cyst, but they do not appear to exhibit the degree of scattering observed in *PhLP3* mutants [74]. Only mutants in the *Clathrin heavy chain* gene exhibit infrequent individualization complex formation and nuclear scattering, similar to our mutants [74]. While numerous genes have been identified to regulate individualization, including regulators of the actin and microtubule cytoskeleton (reviewed in [77]), the signal or event that initiates this process remains unknown.

In mammals, the proper localization of spermatids appears to be an important precursor step to the final steps of nuclear elongation and individualization. An adherens junction-like adhesion complex, known as the ectoplasmic specialization, forms between the Sertoli cell and the spermatid to promote the positioning of the spermatid as it continues sperm maturation (reviewed in [78, 79]). Actin and some adherens junction components have been reported to accumulate at the tip of the nucleus in *D. melanogaster*, suggesting a similar mechanism could function to align spermatids prior to the start of individualization [80]. However, the effect of reducing levels of these components in the germline and surrounding somatic cyst cells has not been examined. As *PhLP3* mutants begin nuclear elongation but do not complete the process and begin individualization, the arrest of spermatid maturation may occur as a result of the failure of spermatid nuclei to align properly.

In this study, we identified the *D. melanogaster* PhLP3 protein and demonstrated its redox activity. We have shown that PhLP3 functions in the late stages of sperm development based on arrested spermatid maturation during the canoe stages in *PhLP3* mutants. Nuclei fail to align and elongate and are scattered through the distal end of the testis. In addition, mutant seminal vesicles are smaller in size when compared to controls and lack mature sperm, resulting in male sterility (Fig 11). *PhLP3-RNAi* results in slightly milder defects, with some sperm

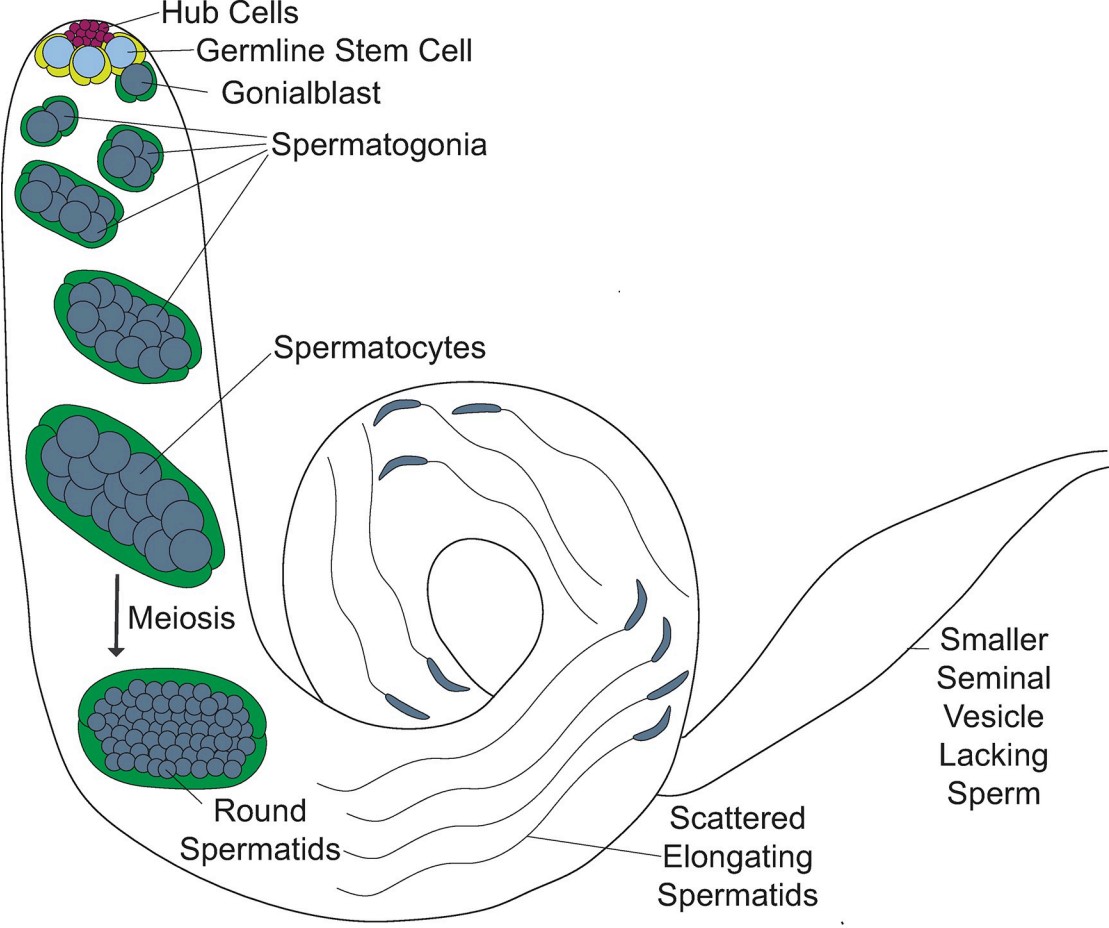

**Fig 11. Model of defects observed in *PhLP3* mutants.** Spermatid nuclei are misshapen and scattered, failing to reach the needle-like stage. Individualization complexes are not observed in *PhLP3* mutants and sperm do not transfer to the seminal vesicle, leading to reduced seminal vesicle size.

still clustering but still failing to complete elongation and to be transferred to the seminal vesicle. It is interesting to note that our RNAi flies with 23% PhLP3 expression compared to controls exhibit significant defects in spermatogenesis, while P element excision flies with 33% PhLP3 expression compared to their respective control exhibit rescue of the mutant phenotype. The different genetic backgrounds of the controls and experimental samples in each experiment may contribute to these observations, resulting in different levels of *PhLP3* expression in each control. It is also possible that our results suggest there may be a critical threshold for *PhLP3* expression that is needed to promote proper sperm development and dropping slightly below that threshold yields significant defects. Further studies are needed to explore this threshold. Our results demonstrate the requirement for PhLP3 in sperm maturation; however, the mechanism by which PhLP3 acts to promote sperm maturation remains unclear. While PhLP proteins have been demonstrated to interact with the CCT to regulate actin and tubulin folding in other species, it could play a different role in this context. Future studies exploring the localization of PhLP3 and its ability to interact with actin, tubulin and CCT will provide further insight into how PhLP3 acts to promote sperm maturation.

## Supporting information

**S1 Fig. *D. melanogaster* possess three PhLPs.** (A) Clustal Omega alignment of the three predicted PhLPs [15, 20]. The conserved residues are indicated with gray boxes. The green boxes indicated the position of the thioredoxin domain with the putative redox-active cysteine highlighted in yellow. The percent identity between each sequence is shown in the box. (B) Comparison of the three PhLPs. Length and characteristic organization of the proteins are indicated. The blue bracket indicates the portion of each protein that was modeled. (C) Hypothetical models of the three predicted PhLP proteins in *D. melanogaster*. The models were generated using SWISS-MODEL template PDB 7nvm: entity 11, human PhLP2A The structures are colored by confidence level from high (blue) to low (red) [17, 34].
(TIF)

**S2 Fig. RNA-seq data for *PhLP3* in dissected tissues.** (A) RNA-Seq expression data from select adult dissected tissues from the modENCODE project. Values listed are Reads per kilobase of transcript per Million reads (RPKM) [21]. (B) RNA-Seq expression data from select adult dissected tissues from the FlyAtlas2 project. Values listed are Reads per kilobase of transcript per Million reads (RPKM) [22]. We used FlyBase (FB2023_06, released December 12, 2023) to obtain the data presented in this figure. (C) Single-cell RNA-Seq expression data in the adult testis from the Fly Cell Atlas project is displayed on Uniform Manifold Approximation and Project (UMAP) graph [23]. In the left panel, shades of red indicate expression levels with brightest red being the highest expression and black being low expression. The UMAP in the right panel illustrates the broadly classified different cell clusters in the testis. *PhLP3* is most highly expressed in the male germline cells. UMAPs were generated in SCope [81].
(TIF)

**S3 Fig. Schematic of *PhLP3* gene region and analysis of P element insertion site.** (A) Schematic of the *PhLP3* gene region. The location of the P element insertion P{EPgy2}$^{EY13373}$ is indicated 78 bp upstream of the translation start site in the 5' UTR. (B) Sequence around the insertion site of P element. The following sequences and sequence data are represented: genomic sequence from FlyBase (FBgn0037843), sequencing result of the control strain ($w^{1118}$), sequencing result of the *PhLP3*$^{\Delta P/\Delta P}$ excision strain, inverse PCR (iPCR) sequencing result of the P element insertion strain (*PhLP3*$^{EY13373}$), and the previously reported flanking sequence for the P element. Sites, where HpaII would cut the sequence, are indicated, as this enzyme

was used for iPCR. The sequenced regions of the P element insertion strain that do not align with the consensus sequence corresponding to the sequence from the P element, P{EPgy2}, not the *PhLP3* gene.
(TIF)

**S1 Table. List of primers used in this study.**
(DOCX)

**S1 Raw image. SDS page electrophoresis following recombinant *Drosophila melanogaster* PhLP3 purification over a Ni-NTA column.**
(TIF)

## Acknowledgments

Thank you to the Bloomington *Drosophila* Stock Center and the Vienna *Drosophila* Resource Center for fly stocks. The rat anti-Vasa monoclonal antibody, developed by A. C. Spradling and D. Williams, was obtained from the Developmental Studies Hybridoma Bank, created by the NICHD of the NIH and maintained at The University of Iowa, Department of Biology, Iowa City, IA 52242. Thanks to E. O'Flaherty and Danielle Talbot for helpful discussions and comments on this project.

## Author Contributions

**Conceptualization:** Jennifer C. Jemc, Stefan M. Kanzok.

**Data curation:** Christopher Petit, Elizabeth Kojak, Samantha Webster, Michela Marra, Brendan Sweeney, Claire Chaikin, Jennifer C. Jemc, Stefan M. Kanzok.

**Formal analysis:** Christopher Petit, Elizabeth Kojak, Samantha Webster, Michela Marra, Brendan Sweeney, Claire Chaikin, Jennifer C. Jemc, Stefan M. Kanzok.

**Funding acquisition:** Jennifer C. Jemc, Stefan M. Kanzok.

**Investigation:** Jennifer C. Jemc, Stefan M. Kanzok.

**Methodology:** Christopher Petit, Jennifer C. Jemc, Stefan M. Kanzok.

**Project administration:** Jennifer C. Jemc, Stefan M. Kanzok.

**Resources:** Jennifer C. Jemc, Stefan M. Kanzok.

**Software:** Jennifer C. Jemc, Stefan M. Kanzok.

**Supervision:** Jennifer C. Jemc, Stefan M. Kanzok.

**Validation:** Michela Marra, Claire Chaikin, Jennifer C. Jemc, Stefan M. Kanzok.

**Visualization:** Christopher Petit, Samantha Webster, Michela Marra, Brendan Sweeney, Jennifer C. Jemc, Stefan M. Kanzok.

**Writing – original draft:** Jennifer C. Jemc, Stefan M. Kanzok.

**Writing – review & editing:** Christopher Petit, Samantha Webster, Michela Marra, Claire Chaikin, Jennifer C. Jemc, Stefan M. Kanzok.

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
