## [Decision Letter · Decision Letter 0]

23 Jul 2024

PONE-D-24-25002The evolutionarily conserved PhLP3 is essential for sperm development in Drosophila melanogasterPLOS ONE

Dear Dr. Kanzok,

Thank you for submitting your manuscript to PLOS ONE. After careful consideration, we feel that it has merit but does not fully meet PLOS ONE’s publication criteria as it currently stands. Therefore, we invite you to submit a revised version of the manuscript that addresses the points raised during the review process.

**The reviews for the above manuscript have been received. The reviewers' comments are included along with this letter.**

**Independent reviewers have assessed the manuscript's suitability for publication and solicit additional clarification on certain outstanding issues.**

**Based on the reports of the reviewers and my own assessment, I recommend that you please revise the manuscript to address the concerns raised.**

**If you wish to revise the manuscript to address the issues raised by reviewers, editors would be happy to consider the revised version of the manuscript for publication.**

**When revising the manuscript, please consider all the points raised by the reviewers and outline every change made. If you disagree with the reviewers' comments, please provide a suitable rebuttal to their concerns.**

We look forward to receiving your revised manuscript.

Kind regards,

Abhinava Kumar Mishra, PhD

Academic Editor

PLOS ONE

Journal Requirements:

This work was made possible by NSF MRI #1828164, which provided for the purchase of the Zeiss LSM880 confocal microscope. This work was supported by internal research awards to C. P., E. K., S. W., M. M., B. S., C. C., J. C. J., and S. M. K. from Loyola University Chicago.

4. Please expand the acronym “C. P., E. K., S. W., M. M., B. S., C. C., J. C. J., and S. M. K.” (as indicated in your financial disclosure) so that it states the name of your funders in full.

Thank you to the Bloomington Drosophila Stock Center and the Vienna Drosophila Resource Center for fly stocks. The rat anti-Vasa monoclonal antibody, developed by A. C. Spradling and D. Williams, was obtained from the Developmental Studies Hybridoma Bank, created by the NICHD of the NIH and maintained at The University of Iowa, Department of Biology, Iowa City, IA 52242. Thanks to E. O’Flaherty and Danielle Talbot for helpful discussions and comments on this project. This work was made possible by NSF MRI #1828164, which provided for the purchase of the Zeiss LSM880 confocal microscope. This work was supported by internal research awards to C. P., E. K., S. W., M. M., B. S., C. C., J. C. J., and S. M. K. from Loyola University Chicago.

This work was made possible by NSF MRI #1828164, which provided for the purchase of the Zeiss LSM880 confocal microscope. This work was supported by internal research awards to C. P., E. K., S. W., M. M., B. S., C. C., J. C. J., and S. M. K. from Loyola University Chicago.

7. PLOS ONE now requires that authors provide the original uncropped and unadjusted images underlying all blot or gel results reported in a submission’s figures or Supporting Information files. This policy and the journal’s other requirements for blot/gel reporting and figure preparation are described in detail at https://journals.plos.org/plosone/s/figures#loc-blot-and-gel-reporting-requirements and https://journals.plos.org/plosone/s/figures#loc-preparing-figures-from-image-files. When you submit your revised manuscript, please ensure that your figures adhere fully to these guidelines and provide the original underlying images for all blot or gel data reported in your submission. See the following link for instructions on providing the original image data: https://journals.plos.org/plosone/s/figures#loc-original-images-for-blots-and-gels.   

Reviewers' comments:

Reviewer's Responses to Questions

**Comments to the Author**

1. Is the manuscript technically sound, and do the data support the conclusions?

Reviewer #1: Yes

Reviewer #2: Partly

Reviewer #3: Yes

Reviewer #4: Yes

2. Has the statistical analysis been performed appropriately and rigorously? 

Reviewer #1: Yes

Reviewer #2: Yes

Reviewer #3: Yes

Reviewer #4: Yes

3. Have the authors made all data underlying the findings in their manuscript fully available?

Reviewer #1: Yes

Reviewer #2: Yes

Reviewer #3: Yes

Reviewer #4: Yes

4. Is the manuscript presented in an intelligible fashion and written in standard English?

Reviewer #1: Yes

Reviewer #2: Yes

Reviewer #3: Yes

Reviewer #4: Yes

5. Review Comments to the Author

**Reviewer #1:** This manuscript presents a strong initial characterization of Drosophila melanogaster PhLP3. The authors use structural modeling and alignment to demonstrate the protein’s conserved structure with PhLP3 homologs from other species, run a biochemical assay to show its redox activity, and then use two complementary genetic approaches (a P-element insertion allele and RNA interference) to establish that ablation of PhLP3 results in male sterility. This is accompanied by cytological studies that identify a lack of individualization complexes in spermatids, an interesting observation given the potential for PhLP3 to interact with cytoskeleton components. This is a very solid manuscript overall. It makes good use of a wide variety of techniques and approaches and will make a helpful contribution to the literature on phosducin-like proteins and on fly spermatogenesis. I have only minor comments, which I expect can be addressed without any additional experiments. (Some of the comments are essentially copy-editing, which the instructions ask reviewers to do, since the journal does not provide this service.)

The excision of the P-element to rescue the wild-type phenotype is a strength and helps validate the P-insertion allele as the cause of the fertility defect. The use of RNAi to confirm the requirement of PhLP3 for fertility is another important strength, making it less likely that the P-element mutant effect was due to some other mutation in that background.'

Minor points:

Figure 1:

-Give units for the distance measure included at the top of the phylogenetic tree in Fig. 1B.

-Specify in the caption whether the structure modeled in Fig. 1C is the Drosophila PhLP3, or another ortholog.

-If you haven’t already, consider comparing the SWISS-MODEL structure to the AlphaFold prediction (https://alphafold.ebi.ac.uk/entry/Q9VGV8) to determine which is the better one to include in the figure. To my non-expert eye, they appear fairly similar, though the AlphaFold prediction shows a longer second alpha helix at the N-terminus (in line with the discussion section description, line 549). If you feel SWISS-MODEL is better, then no need to change. If you think the AlphaFold prediction is more reliable, then you could do the same for the structures in Fig. S1.

-If any amino acids are missing from the structure, you could clarify this in the caption (e.g., it looks like positions 1-18 and 185-216 are not represented?).

-If possible, remove Microsoft’s red-squiggly line under Trx.

Line 114: The data in Fig. S2 do not seem to distinguish between “germline” and somatic cells of the testis, so consider re-wording this sentence (or, clarify that germline-specific expression is based on scRNA-seq data and/or your later data in Fig. 6, rather than the graphs in the supplemental figure?).

Figure 2:

-Add to the caption that “dc” indicates the dense complex in Fig. 2B.

Line 151: no italics for “mutants”

Figure 3:

-If possible, remove Microsoft’s red-squiggly line under Trx.

Figure 4:

-Fig. 1 also presents a PhLP3 structure and an alignment, so aspects of these figures are redundant with each other. Since there doesn’t appear to be anything anomalous in the alignments or structures that is noteworthy for the manuscript (i.e., the structures align well, the cysteine is conserved, etc.), can these figures be condensed into one, with any other information being moved to supplemental material?

-In the caption describing Fig. 4A, clarify which structure is shown in part with the green curves.

-Line 190: unnecessary underlining of ref. [16].

Line 200: it may be helpful to give a brief (one line) description of “the thioredoxin system” here, since this system might be unfamiliar to some readers, or simply add “(describe in the next paragraph)”, where it’s clarified.

Line 202: Do you mean Cys95 in D. melanogaster, rather than Cys93? (Also, be consistent with how you write the amino acids; cysteine is abbreviated as C in line 195, Cys here.)

Figure 5:

-It may be helpful to show the chemical reaction in Fig. 5B with the same protein names as you use in the text. For example, in line 215, you write about D. melanogaster “Trx1”, but it’s just “Trx” in Fig. 5B. Overall, though, these look like great data!

Figure 6:

-This is a nice demonstration that the P-element insertion abolishes PhLP3 expression.

-The first line of the figure caption (line 243) states that panels “(A-B)” show in situ data, but it looks like only panels A and C show this, while panel B shows qPCR data. Please clarify.

Figure 8:

-These data are convincing that PhLP3-/- males do not produce mature sperm and do not undergo individualization.

-To support the text on lines 322-323, is it possible to show images with only the phalloidin channel here, to illustrate the presence of ICs for controls and the absence of them for the mutants? (If not, that is okay; this is a small point.)

Figure 9:

-Is it possible to add to the caption whether these images represent zoomed in portions of whole-mount (intact) testes (which seems to be what the Methods section implies on line 450), or whether they come from shreds to release spermatid bundles?

It is a bit surprising that the P-element excision line (with 33% of WT levels of PhLP3 transcripts) is fertile, while the RNAi line (with 23% of WT levels of PhLP3 transcripts) is near-sterile. Either the authors have serendipitously narrowed down the exact threshold level of PhLP3 transcripts required for functional spermatogenesis, or there are other biological or technical explanations. (Example biological: there could be post-transcriptional regulation that acts differently between the delta P line and the RNAi progeny or some strain background differences with the RNAi line. Example technical: there could be variation in the RT-PCR measurements. The bam-gal4 control line measurement in Fig. 10 has large error bars, for instance, while error bars appear to be missing for the +/+ genotype in Fig. 6.) This doesn’t require extensive comment, but if the authors had thoughts about it, they could briefly include them.

Figure 10:

-Optional: if possible, it would be useful to show the scattered actin cones in the RNAi flies at higher magnification, but in my view, it is not necessary to re-do this experiment if you don’t already have the data available.

Line 419: “or control” doesn’t need to be in italics

Line 474-475: it’s not clear why Trx1 primers were used, since the manuscript doesn’t report on Trx1 purification. If the authors needed to produce and purify Trx1 for the biochemical assay, this can be briefly added to the manuscript, or they can simply state that Trx1 was produced as previously described in ref. 4.

Given the high expression of CG4511 in ovaries, did the authors observe whether homozygous P-element insertion females were sterile?

Lines 568-569: Related to the point above: This part of the discussion seems to raise the possibility that PhLP3 could be a redox target of Dhd in the ovaries. There doesn’t yet appear to be any evidence that PhLP3 is part of mature sperm (for example, while not definitive evidence for absence, CG4511 was not detected amongst the 3,000+ sperm proteome proteins; Garlovsky et al. 2022, PMID: 35985624), so it isn’t clear that testis-expressed PhLP3 would have the opportunity to interact with Dhd. Alternatively, since the data in Fig. S2 show high PhLP3 expression in ovaries as well as testes, it could be possible that ovary-expressed PhLP3 is a target of Dhd. Thus, consider re-phrasing here.

Line 571-572: Clarify this sentence a bit. What aspect of the kinetics data presented here (and, compared to what other data) suggests that PhLP3 is unlikely to be an effective antioxidant scavenger?

Line 638: remove the extra ) after the parentheses

Line 641: “…described by Fabrizio have … [68].”

**Reviewer #2:** The paper entitled The evolutionarily conserved PhLP3 is essential for sperm development in Drosophila melanogaster provides an analysis of the role and function of CG4511, a conserved member of the phosducin-like family of proteins. In brief, the authors examine the structural similarity of the protein to other known members of the PhLP family, and perform several experiments examining sperm development, noting an arrest of spermatogenesis in PhLP3¬-/- flies. A significant impact on fertility in PhLP3 null animals is noted both P-element excision and RNAi knockdown are used to further interrogate the results.

In accordance with the PLOS ONE publishing criteria, please find my assessment below:

1) The study presents the results of primary scientific research.

a. Yes

2) Results reported have not been published elsewhere.

a. To the best of my knowledge, these results are novel. The entry in flybase would also indicate this CG4511 is understudied.

3) Experiments, statistics, and other analyses are performed to a high technical standard and are described in sufficient detail.

a. Yes.

4) Conclusions are presented in an appropriate fashion and are supported by the data.

a. The format of the article may require some adjustment for clarity, but is sufficient to replicate the experiments.

b. I have significant concern in two areas, given what is reported in the article:

i. In line 349-351, the authors denote that the p-element excision used in the article only restores expression to the level of 33%, but restores fertility in the animals. However, knockdown by RNAi (by 77%, as per line 356-357) is sufficient to render the animals infertile. Based on the p-element data, I would expect the knockdown to not render the animals sterile based on expression. I believe that further explanation is required beyond the expression levels to be confident in the validity of this data.

ii. Only one RNAi line is used in the paper. I would much prefer to see at least two different lines, especially given the potential issue raised prior.

5) The article is presented in an intelligible fashion and is written in standard English.

a. Yes.

6) The research meets all applicable standards for the ethics of experimentation and research integrity.

a. Yes.

7) The article adheres to appropriate reporting guidelines and community standards for data availability.

a. Yes.

**Reviewer #3:** The authors find a phosducin-like (PhLP) protein family gene, CG4511, in Drosophila, which is named PhLP3. The gene is mainly expressed in testes and ovaries according to the published RNAseq data. An insertional mutation in the 5’ UTR caused a significant reduction of expression and the homozygous males are sterile because of a lack of mature sperm. Importantly, the defects are seen as early as the canoe stage, where the microtubule and actin-rich structure, dense body is elongated and leads to needle-shaped nuclear. stage. The authors also confirmed its redox activity by using recombinant protein and NADPH reduction assays.

Because PhLP proteins are known to play a role in regulation of the microtubule and actin cytoskeletons, it is convincing that the mutant of the Drosophila PhLP gene with testis-biased expression showed defective spermiogenesis and sterile. I think that this study finds a new role of PhLP protein in Drosophila spermiogenesis and the paper merits publication in PLOS ONE. However, I found several issues that the authors should address before accepting.

Line 114 “high expression in the testis, specifically in germline cells (S2 Fig)”

I do not see any evidence for expression in germline cells in the figure. Is there any more data?

Line 170 “ an N-terminal helix domain, the central Trx-domain, and the unstructured N-terminal tail”

“unstructured N-terminal tail” is right or unstructured C-terminal tail? This is because “mostly unstructured C-terminal tail” appears on line 543.

Line 239 “gene is expressed in germ cells”

It is difficult to see the exact location of expression in the figure. It may be recommended to add an enlarged view of A.

Figure 7

Nonparametric Welch’s t-test is recommended, particularly between the control and mutant as in Figure 10.

Fig. 8 and line 322

It is difficult to see phalloidin staining (actin cones) in A and also to find sperm in seminal vesicles. Is it possible to add more focused and enlarged view? In addition, it may also be better to combine Figures 8 and 9 in order to understand the phenotypes.

Figure 9

It takes time to understand which are open and solid arrows; alternative symbols are better.

Figure 10

“White arrows indicate clusters of elongating spermatid nuclei” This is phalloidin staining; the revision is needed. There are no “arrows” in C. I cannot see evidence for “there are few if any DAPI clusters” in D or cannot understand where I should see. It is also difficult to see exactly “deformed, scattered actin cones.” Because this is very important point of this paper, the authors should be strongly recommended to prepare double staining and enlarged views for this.

Line 460 “manufacturer’s instructions”

Which is the manufacturer?

Line 549 “Stirling et al.”

Reference number [6] should be just after the author name.

Line 613 “Others”

Reference or references should be provided here.

**Reviewer #4: **Successful completion of spermatogenesis is crucial for the perpetuation of the species. It is interesting for authors used Drosophila as model to investigate gene function in spermatogenesis. The authors uncovered an important role of PhLP3 in spermiogenesis. These results could deepen the basic theory of the regulatory mechanism of animal reproductive development. However, some revisions are required before this paper can be accepted.

Major comments

1. It needs to be clear the reason of male sterile is due to the early release of mature sperm in the experimental group leads to infertility or the absence of spermatid extension stage abnormalities leads to abnormal individualization.

2. PhPL3 is a chaperone protein involved in protein folding as well as a role as an enzyme. Whether the protein degradation pathways were activated in the absence of PhPL3, such as ubiquitination proteasomal degradation pathway, lysosomal pathway, caspase. It is preferable do detect some gene expression related to these pathway.

3. It is preferable do detect some gene expression related to spermatogenesis in the absence of PhPL3.

4. In the result session, authors should explain the reason of using bam-gal4.

Minor comments

5.It is perferable to mark “seminal vesicle” in figure 8 and 10, thus easy for readers.

6.“***” shouled be marked in figure 8A.

6. PLOS authors have the option to publish the peer review history of their article (what does this mean?). If published, this will include your full peer review and any attached files.

Reviewer #1: No

Reviewer #2: No

Reviewer #3: No

Reviewer #4: No

---

## [Author Response · Author response to Decision Letter 0]

6 Sep 2024

RESPONSE TO EDITOR AND REVIEWERS

Dear Editor and Reviewers, 

We sincerely thank you for your time, consideration, and careful review of our manuscript. According to your suggestions, the manuscript has been revised where applicable. Please find our responses to all the issues raised below. Our answers are organized by reviewer number. 

AMENDED STATEMENTS FOR JOURNAL REQUIREMENTS

• We checked and corrected the style requirements according to PLOS ONE requirements.

• We corrected the grant numbers in “Funding information” and “financial disclosure” to NSF MRI #1828164. 

• We updated the Funding statement according to your instructions. 

• Amended statement:

• This work was supported by NSF MRI #1828164, which provided for purchasing the Zeiss LSM880 confocal microscope (JCJ and SMK). This work was furthermore supported by internal research awards from Loyola University Chicago (CP, EK, SW, MM, BS, CC, JCJ, and SMK). The funders had no role in the data collection and analysis, publication decision, or manuscript preparation. No additional external funding was received for this study.

• We entered the author's initials incorrectly, and the sequence “C. P., E. K., S. W., M. M., B. S., C. C., J. C. J., and S. M. K.” was misread as an acronym. 

• We corrected the format for author initials in the amended funding statement to CP, EK, SW, MM, BS, CC, JCJ, and SMK, respectively.

• The acknowledgments statement has been updated following your instructions.

• Amended Acknowledgement statement:

• Thank you to the Bloomington Drosophila Stock Center and the Vienna Drosophila Resource Center for fly stocks. The rat anti-Vasa monoclonal antibody, developed by A. C. Spradling and D. Williams, was obtained from the Developmental Studies Hybridoma Bank at the University of Iowa, Department of Biology, Iowa City, IA 52242. Thanks to E. O’Flaherty and Danielle Talbot for helpful discussions and comments on this project.

• We have updated our data availability statement.

• Updated data sharing statement: 

• The authors confirm that the data supporting the findings of this study are available within the article and its supplemental material.

• An uncropped and unadjusted image of the SDS gel image as well as a corresponding legend has been included in the Supporting Information under the file name S3_Fig.eps.

RESPONSE TO REVIEWERS’ COMMENTS

REVIEWER #1

-Give units for the distance measure included at the top of the phylogenetic tree in Fig. 1B.

• We labeled the scale of the phylogenetic tree and added an explanation to the legend: The scale represents the "length" of the branches, which indicates the evolutionary distance between the sequences in units of amino acid substitutions per site. Longer branches represent larger numbers of genetic changes.

-Specify in the caption whether the structure modeled in Fig. 1C is the Drosophila PhLP3 or another ortholog.

• We have removed the 3D model of Drosophila PhLP3 from Fig 1, as it does not add to the introduction. We have generated a new Fig. 4 focusing on Drosophila PhLP3 structure prediction and modeling (see explanation in the next paragraph). 

-If you haven’t already, consider comparing the SWISS-MODEL structure to the AlphaFold prediction (https://alphafold.ebi.ac.uk/entry/Q9VGV8) to determine which is the better one to include in the figure. To my non-expert eye, they appear fairly similar, though the AlphaFold prediction shows a longer second alpha helix at the N-terminus (in line with the discussion section description, line 549). If you feel SWISS-MODEL is better, then no need to change. If you think the AlphaFold prediction is more reliable, then you could do the same for the structures in Fig. S1.

• Thank you for your suggestion. We have considered the AlphaFold prediction, which coincidentally is the first template suggestion in SWISS-MODEL. We agree that the AlphaFold prediction shows the entire structure of Drosophila PhLP3, most notably the N-terminal helices and the thioredoxin domain, with a high confidence index. We generated a new Figure 4 that compares the AlphaFold and SWISS-MODEL structures of Drosophila PhLP3. We included the SWISS-MODEL shown in Fig 4B, as it is based on the experimentally determined structure of human PhLP2A. Notably, the template file with the PDB code 7NVM contains the CryoEM structure of the entire chaperonin TRiC/CCT complexed with PhLP2A and actin inside its luminal catalytic cavity. We hypothesize that Drosophila PhLP3 interactions with Drosophila TRiC/CCT are similar. Even though the PhLP2A structure is truncated, and consequently our Drosophila PhLP3 model too, it represents the structure under functional, cellular conditions. 

-If any amino acids are missing from the structure, you could clarify this in the caption (e.g., it looks like positions 1-18 and 185-216 are not represented?).

• In the new Fig. 4A (AlphaFold prediction), we highlighted the N-terminal and C-terminal sections missing in Fig. 4B (SWISS-MODEL). We explained the missing section in the main text (lines 179 – 181): Notably, the first 18 N-terminal AA and the last 25 AA at the C-terminus were not modeled, as they were unresolved in the hPhLP2A crystal structure within the chaperonin complex. 

Line 114: The data in Fig. S2 do not seem to distinguish between “germline” and somatic cells of the testis, so consider re-wording this sentence (or, clarify that germline-specific expression is based on scRNA-seq data and/or your later data in Fig. 6, rather than the graphs in the supplemental figure?).

• Line 114 (now line 115) We have added panel C to Fig. S2, which includes the scRNA-Seq data that illustrates PhLP3 expression in the germline from Li et al., 2022. 

Figure 2:

-Add to the caption that “dc” indicates the dense complex in Fig. 2B.

Line 151: no italics for “mutants”

• Figure 2: We have edited the Figure 2B legend, adding “dc: dense complex.”

Line 151: no italics for “mutants”

• Line 151 (now line 153) “mutants” is no longer italicized.

Figure 3:

-If possible, remove Microsoft’s red-squiggly line under Trx.

• Figure 3: The squiggly lines under Trx have been removed.

Figure 4:

-Fig. 1 also presents a PhLP3 structure and an alignment, so aspects of these figures are redundant with each other. Since there doesn’t appear to be anything anomalous in the alignments or structures that is noteworthy for the manuscript (i.e., the structures align well, the cysteine is conserved, etc.), can these figures be condensed into one, with any other information being moved to supplemental material?

• Figure 4 (new) Thank you. Following your suggestion, we have removed the SWISS-MODEL from Fig. 1 and generated a new Fig. 4 that shows the AlphaFold prediction as well as the SWISS-MODEL of D. melanogaster PhLP3. Fig. 1 A and B relate to the introduction while the modeling was performed in this study and is, therefore, part of the results section. We moved all structures to Figure 4. 

-In the caption describing Fig. 4A, clarify which structure is shown in part with the green curves.

• Figure 4A (now 4B): We added the following explanation to the legend of our new Fig 4B: “The green portions of the PhLP3 structure represent unstructured segments, including loops and turns.”

-Line 190: unnecessary underlining of ref. [16].

• Line 190 (now 197): ref [16] is now properly formatted and listed as ref [32] and is no longer underlined. 

Line 200: it may be helpful to give a brief (one line) description of “the thioredoxin system” here, since this system might be unfamiliar to some readers, or simply add “(describe in the next paragraph)”, where it’s clarified.

• Line 200 (now line 210): We have included the following sentence at the beginning of that paragraph. “The cytosolic thioredoxin system is central in distributing electrons to target proteins in various antioxidant and regulatory pathways.” 

Line 202: Do you mean Cys95 in D. melanogaster, rather than Cys93? (Also, be consistent with how you write the amino acids; cysteine is abbreviated as C in line 195, Cys here.)

• Line 202 (now line 215): We corrected CYS93 to CYS95 and used this format throughout the manuscript.

Figure 5:

-It may be helpful to show the chemical reaction in Fig. 5B with the same protein names as you use in the text. For example, in line 215, you write about D. melanogaster “Trx1”, but it’s just “Trx” in Fig. 5B. Overall, these look like great data!

• Figure 5: As CG4193 is annotated as DmTrx-1 in FlyBase, we therefore adjusted Trx and Trx1 annotations to Trx-1 or DmTrx-1 throughout the manuscript where appropriate and for consistency, including Fig. 5B. 

Figure 6:

-The first line of the figure caption (line 243) states that panels “(A-B)” show in situ data, but it looks like only panels A and C show this, while panel B shows qPCR data. Please clarify.

• Figure 6: We updated the labeling within the Figure 6 legend (now line 256) and the figure description in the text to match. A new panel was added to show the germline staining with anti-Vasa alone compared to the PhLP3 in situ to better demonstrate PhLP3 expression in the germline.

Figure 8:

-To support the text on lines 322-323, is it possible to show images with only the phalloidin channel here, to illustrate the presence of ICs for controls and the absence of them for the mutants? (If not, that is okay; this is a small point.)

• Figure 8: Text lines 322-323 (now 344-345): We added new sets of panels showing only the phalloidin channel.

Figure 9:

-Is it possible to add to the caption whether these images represent zoomed in portions of whole-mount (intact) testes (which seems to be what the Methods section implies on line 450), or whether they come from shreds to release spermatid bundles?

• Figure 9: We modified the caption to specify that these images were taken from whole mount testes.

It is a bit surprising that the P-element excision line (with 33% of WT levels of PhLP3 transcripts) is fertile, while the RNAi line (with 23% of WT levels of PhLP3 transcripts) is near-sterile. Either the authors have serendipitously narrowed down the exact threshold level of PhLP3 transcripts required for functional spermatogenesis, or there are other biological or technical explanations. (Example biological: there could be post-transcriptional regulation that acts differently between the delta P line and the RNAi progeny or some strain background differences with the RNAi line. Example technical: there could be variation in the RT-PCR measurements. The bam-gal4 control line measurement in Fig. 10 has large error bars, for instance, while error bars appear to be missing for the +/+ genotype in Fig. 6.) This doesn’t require extensive comment, but if the authors had thoughts about it, they could briefly include them.

• The reviewer describes the exciting observation that flies with just 33% of PhLP3 expression (P element excision line) relative to controls produce sperm and are fertile. In comparison, those with 23% of PhLP3 expression (RNAi line) relative to controls are sterile and fail to produce mature sperm. As suggested by the reviewer, it is possible that our results indicate a critical threshold for PhLP3 expression required for protein function during sperm development. It is notable that the controls for each of these experiments are in different genetic backgrounds, and therefore, the levels across controls may not be the same. We have added some discussion about this observation and these possibilities to the final paragraph of the discussion section.

Figure 10:

-Optional: if possible, it would be useful to show the scattered actin cones in the RNAi flies at higher magnification, but in my view, it is not necessary to re-do this experiment if you don’t already have the data available.

• Fig. 10: We added 20X pictures (panels H and N) to illustrate the RNAi phenotype and the actin cones.

Line 419: “or control” doesn’t need to be in italics

• Line 419 (now line 442): Removed italics of “or control.”

Line 474-475: it’s not clear why Trx1 primers were used, since the manuscript doesn’t report on Trx1 purification. If the authors needed to produce and purify Trx1 for the biochemical assay, this can be briefly added to the manuscript, or they can simply state that Trx1 was produced as previously described in ref. 4.

• Line 474-475 (now 497-498): We removed the DmTrx-1 primers from Table S1 and added the suggested statement in the Enzyme assays section (line 512-513) “D. melanogaster thioredoxin-1 (DmTrx-1, CG4193, [37]) was produced as described previously [4,35]”. We added another reference [35] (Kanzok et al. 2001); we corrected the naming of Trx to Trx-1 for consistency, as requested earlier. 

Given the high expression of CG4511 in ovaries, did the authors observe whether homozygous P-element insertion females were sterile?

• We appreciate the reviewer's question about the fertility of homozygous P-element insertion females. We have observed that these females are sterile, and we are currently studying the effect of PhLP3 mutation on oogenesis. We plan to publish this work soon when we have more information on the role of PhLP3 in females. Therefore, we chose not to include the data here.

Lines 568-569: Related to the point above: This part of the discussion seems to raise the possibility that PhLP3 could be a redox target of Dhd in the ovaries. There doesn’t yet appear to be any evidence that PhLP3 is part of mature sperm (for example, while not definitive evidence for absence, CG4511 was not detected amongst the 3,000+ sperm proteome proteins; Garlovsky et al. 2022, PMID: 35985624), so it isn’t clear that testis-expressed PhLP3 would have the opportunity to interact with Dhd. Alternatively, since the data in Fig. S2 show high PhLP3 expression in ovaries as well as testes, it could be possible that ovary-expressed PhLP3 is a target of Dhd. Thus, consider rephrasing here.

• Lines 568-569 (now lines 594-595): The reviewer mentions the lack of detection of CG4511 among the sperm proteome proteins in Garlovsky et al., 2022. This study utilizes sperm isolated from the seminal vesicle for these proteome studies. Our data suggests that CG4511/PhLP3 functions prior to sperm reaching maturity and being transferred to the seminal vesicle. It is possible that PhLP3 may no longer be expressed and required in mature sperm. Interestingly, Dhd is highly expressed in male germline cells of the testis, based on scRNA-seq data. Considering the ubiquitous presence of Dhd, this supports the hypothesis of an interaction between PhLP3 and Dhd.

Line 571-572: Clarify this sentence a bit. What aspect of the kinetics data presented here (and, compared to what other data) suggests that PhLP3 is unlikely to be an effective antioxidant scavenger?

• Lines 571-572 (now lines 597-606): We thank the reviewer for pointing this out. The original sentence suggests that we tested the antioxidant activity of Drosophila PhLP3, which we did not. We corrected the statement as follows: The thioredoxin system is a central electron distribution system. Some of its target proteins are thioredoxin-like proteins with strong antioxidant activity [56]. We previously demonstrated that P. berghei PhLP3 and its homolog human TXNDC9 exhibit antioxidant activity that is orders of magnitude weaker than that of antioxidant effectors, such as thioredoxin-dependent peroxiredoxin (TPx1) [4]. We concluded that PhLP3 is unlikely to act as an antioxidant scavenger in the cell. In the fruit fly effective antioxidant systems, such as catalase and superoxide dismutase, cooperate in the fly to protect from reactive oxygen species [57]. The hypothesis for a different function of PhLP3 is corroborated by the recent structural insights into the interactions of PhLP3 and PhLP2A with the CCT complex and its potential role in the folding of actin and tubulin [6,17].

Line 638: remove the extra ) after the parentheses

• Line 638 (Now line 668): extra parenthesis removed

Line 641: “…described by Fabrizio have … [68].”

• Line 641 (now line 671): removed “1998”, as this detail is evident with the citation [68]

REVIEWER#2:

i. In line 349-351, the authors denote that the p-element 

---

## [Decision Letter · Decision Letter 1]

29 Sep 2024

The evolutionarily conserved PhLP3 is essential for sperm development in Drosophila melanogaster

PONE-D-24-25002R1

Dear Dr. Kanzok,

We’re pleased to inform you that your manuscript has been judged scientifically suitable for publication and will be formally accepted for publication once it meets all outstanding technical requirements.

Kind regards,

Abhinava Kumar Mishra, PhD

Academic Editor

PLOS ONE

Additional Editor Comments (optional):

Reviewers' comments:

Reviewer's Responses to Questions

**Comments to the Author**

1. If the authors have adequately addressed your comments raised in a previous round of review and you feel that this manuscript is now acceptable for publication, you may indicate that here to bypass the “Comments to the Author” section, enter your conflict of interest statement in the “Confidential to Editor” section, and submit your "Accept" recommendation.

Reviewer #1: All comments have been addressed

Reviewer #2: (No Response)

Reviewer #3: All comments have been addressed

Reviewer #4: All comments have been addressed

2. Is the manuscript technically sound, and do the data support the conclusions?

Reviewer #1: Yes

Reviewer #2: Yes

Reviewer #3: Yes

Reviewer #4: Yes

3. Has the statistical analysis been performed appropriately and rigorously? 

Reviewer #1: Yes

Reviewer #2: Yes

Reviewer #3: Yes

Reviewer #4: Yes

4. Have the authors made all data underlying the findings in their manuscript fully available?

Reviewer #1: Yes

Reviewer #2: Yes

Reviewer #3: Yes

Reviewer #4: Yes

5. Is the manuscript presented in an intelligible fashion and written in standard English?

Reviewer #1: Yes

Reviewer #2: Yes

Reviewer #3: Yes

Reviewer #4: Yes

6. Review Comments to the Author

Reviewer #1: I appreciate the authors’ careful attention to my comments and those of the other reviewers. The approach of moving the structures to the new Fig. 4 looks to have worked effectively. I also appreciated the additional discussion of the threshold issue, but agree with the authors that it can be explored experimentally in future studies. Likewise, it’s cool that PhLP3 mutant females are sterile, but very reasonable to save that story for another paper. Overall, I think this revised submission looks good and will make an interesting contribution to the literature on fly spermatogenesis.

Reviewer #2: The authors have clarified the majority of my concerns, and the re-write of the paper is significantly improved. Based on the comments added, I would accept it for publication.

However, it may be possible to further clarify both the threshold level of PhLP3 transcripts required for

functional spermatogenesis (as reviewer #1 and I noted) and my concerns about the RNAi experiments by adjusting the temperature for the Gal4 driven RNAi experiments (as another line is not available). This would help to remove any concerns about a background effect or other potential off-target effects.

Reviewer #3: (No Response)

Reviewer #4: (No Response)

7. PLOS authors have the option to publish the peer review history of their article (what does this mean?). If published, this will include your full peer review and any attached files.

Reviewer #1: No

Reviewer #2: No

Reviewer #3: No

Reviewer #4: No

---

## [Editor Report · Acceptance letter]

23 Oct 2024

PONE-D-24-25002R1 

PLOS ONE

Dear Dr. Kanzok, 

I'm pleased to inform you that your manuscript has been deemed suitable for publication in PLOS ONE. Congratulations! Your manuscript is now being handed over to our production team.

Kind regards, 

on behalf of

Dr. Abhinava Kumar Mishra 

Academic Editor

PLOS ONE